# Targeting glutamine metabolism improves sarcoma response to radiation therapy in vivo
Rutulkumar Patel [1,8], Daniel E. Cooper[2,8], Kushal T. Kadakia[2], Annamarie Allen[3], Likun Duan[3,4], Lixia Luo[2], Nerissa T. Williams[2], Xiaojing Liu [4], Jason W. Locasale[3,4] & David G. Kirsch [2,3,5,6,7] ✉

Diverse tumor metabolic phenotypes are influenced by the environment and genetic lesions. Whether these phenotypes extend to rhabdomyosarcoma (RMS) and how they might be leveraged to design new therapeutic approaches remains an open question. Thus, we utilized a *Pax7*$^{Cre-ER-T2/+}$*; Nras*$^{LSL-G12D/+}$*; p53*$^{fl/fl}$ (P7NP) murine model of sarcoma with mutations that most frequently occur in human embryonal RMS. To study metabolism, we infuse $^{13}$C-labeled glucose or glutamine into mice with sarcomas and show that sarcomas consume more glucose and glutamine than healthy muscle tissue. However, we reveal a marked shift from glucose consumption to glutamine metabolism after radiation therapy (RT). In addition, we show that inhibiting glutamine, either through genetic deletion of glutaminase (*Gls1*) or through pharmacological inhibition of glutaminase, leads to significant radiosensitization in vivo. This causes a significant increase in overall survival for mice with *Gls1*-deficient compared to *Gls1*-proficient sarcomas. Finally, *Gls1*-deficient sarcomas post-RT elevate levels of proteins involved in natural killer cell and interferon alpha/gamma responses, suggesting a possible role of innate immunity in the radiosensitization of *Gls1*-deficient sarcomas. Thus, our results indicate that glutamine contributes to radiation response in a mouse model of RMS.

Sarcomas are a heterogeneous group of > 70 distinct cancer subtypes derived from mesenchymal tissues, including bone, adipose, and muscle[1,2]. These tumors account for nearly 20% of all cancer-related deaths in children and adolescents[2,3]. The most commonly diagnosed form of soft tissue sarcoma in children is rhabdomyosarcoma[4]. In embryonal rhabdomyosarcoma (eRMS), the most common recurrent mutations are in the tumor suppressor *p53* and *RAS* genes, most frequently in *NRAS*[5]. Because RAS activity drives an altered metabolism[6] and muscle is an essential metabolic tissue, altered metabolism in muscle-derived and RAS-driven sarcomas may identify targetable metabolic pathways to limit sarcoma growth.

Targeting the metabolism of key nutrients via pharmacological inhibition or dietary restriction can inhibit tumor growth[7–11]. However, given the redundancy of metabolic pathways, blocking a single nutrient may only produce temporary and unsustainable growth restrictions. In some instances, it is possible that these nutrients, per se, may not be required to promote tumor growth on their own, but the by-products of their metabolism may instead contribute an essential component to tumor maintenance. Besides contributing to bioenergetic and biosynthetic pathways, cellular metabolism also supports proliferation by generating essential cofactors and influencing epigenetic changes[12,13]. Recently, progress has been made in understanding sarcoma metabolism. In patients with osteosarcoma, which is the most common bone tumor, serum, and urinary metabolomics revealed a distinct phenotype compared to healthy patients, suggesting that downregulation of central carbon metabolism and increased glutathione metabolism were characteristic features of osteosarcoma[14]. In osteosarcoma cell lines, central carbon metabolism, glutathione

[1]Department of Radiation Oncology, Baylor College of Medicine, 7200 Cambridge St, Houston, TX 77030, USA. [2]Department of Radiation Oncology, Duke University, Box 3085, Duke Cancer Center, Medicine Circle, Durham, NC 27710, USA. [3]Department of Pharmacology and Cancer Biology, Duke University, Box 3813, 308 Research Drive, Durham, NC 27710, USA. [4]Department of Molecular and Structural Biochemistry, NC State University, Box 7622, 128 Polk Hall, Raleigh, NC 27695, USA. [5]Present address: Radiation Medicine Program, Princess Margaret Cancer Centre, University Health Network, 610 University Avenue, Toronto, ON M5G 2M9, Canada[6]Present address: Department of Radiation Oncology, University of Toronto, 149 College Street, Suite 504, Toronto, ON M5T 1P5, Canada[7]Present address: Department of Medical Biophysics, University of Toronto, 101 College Street, Room 15-701, Toronto, ON M5G 1L7, Canada. [8]These authors contributed equally: Rutulkumar Patel, Daniel E. Cooper. ✉e-mail: David.Kirsch@uhn.ca

metabolism, the pentose phosphate pathway, and serine/glycine metabolism were upregulated[15–17]. Furthermore, glutamine dependence of undifferentiated pleomorphic sarcomas has been identified and successfully targeted in vitro and in a syngeneic mouse model to inhibit sarcoma cell growth[8].

Finally, radiation therapy (RT) plays a vital role in the standard of care regimen for many patients with sarcoma. However, radioresistance in high-grade sarcoma can lead to recurrence. Previous studies suggested that RT rewires cancer metabolism through an altered glycolysis[18] and pentose phosphate pathway[19] to deal with radiation-induced hypoxia and DNA damage, respectively. In addition, RT also impacts redox metabolism in cancer via the formation of reactive oxygen species leading to radiation-induced oxidative stress[20]. However, the majority of metabolic alterations were identified in cancer cells using in vitro or cell-transplanted animal models. Although in vitro models have provided valuable insights into the active metabolic networks of cancer cells, the metabolism of cells in culture is often different from the metabolism of tumors in vivo[21], and metabolite profiling albeit informative, does not directly probe metabolic flux and thus incompletely characterizes metabolic pathway activity. Recent studies suggest that the tumor microenvironment plays a significant role in determining the type of fuel that tumors use to grow. For example, experiments using typical in vitro culture conditions of primary lung adenocarcinoma cell lines suggested that glutamine is a predominant anaplerotic substrate for the TCA cycle. However, glutamine anaplerosis was virtually nonexistent when these cell lines were transplanted to the lung or when primary lung cancers were initiated in vivo. Instead, in vivo lung tumors use glucose-derived pyruvate as an anaplerotic substrate via the enzyme pyruvate carboxylase (PC)[21,22]. Furthermore, in 4T1 breast tumors, PC-dependent anaplerosis was negligible, but when these tumors metastasized to the lung, PC-dependent anaplerosis increased dramatically[23]. These findings demonstrate that metabolic pathways supporting tumor growth highly depend on the tumor microenvironment in which the tumors grow and emphasize the significance of studying metabolism with in vivo primary models. To address these limitations and investigate the predominant metabolic alterations occurring post-RT, we performed metabolomics and in vivo nutrient tracing in a genetically engineered mouse model of oncogenic *Nras* and *Trp53* deficient soft tissue sarcoma[24].

## Results
### Disruption of a broad range of metabolites in autochthonous soft-tissue sarcomas

In order to better understand the metabolic adaptations that enable the outgrowth of soft-tissue sarcomas (STS), we utilized a genetically engineered mouse model that mimics rhabdomyosarcoma (RMS) that we previously characterized whereby intramuscular injection of 4-hydroxytamoxifen activated a muscle satellite cell-specific CreER[T2] to turn on expression of an oncogenic *Nras[G12D]* allele and delete both *p53* alleles[24], resulting in local tumor formation (P7NP sarcoma) within 4–6 weeks (Fig. 1a). Once tumors reached ~250mm[3], we collected the sarcomas and the surrounding healthy muscle tissues and performed liquid chromatography/mass spectrometry (LC/MS). A heatmap of the identified metabolites revealed significant differences between sarcoma and muscle tissue (Fig. 1b). The abundance of the majority of metabolites in glycolysis was significantly reduced, while the abundance of the majority of metabolites in the TCA cycle was significantly higher in sarcomas compared to muscle, as shown in Supplementary Fig. 1a, b. In addition, the top six pathways that were different between tumor and muscle were related to nucleotide, amino acid, and glutathione metabolism (Fig. 1c, d). Nucleotides serve as the building blocks for DNA and RNA synthesis, and the elevated nucleotide levels were consistent with the increased need for tumor cell proliferation (Fig. 1e). Arginine and proline-related metabolites were also significantly higher in sarcomas compared to adjacent muscles (Fig. 1f). Most importantly, glutaminolysis was significantly elevated in sarcomas compared to adjacent muscle, suggesting sarcomas' reliance on glutamine metabolism (Fig. 1g). Further, the ratio of oxidized to reduced glutathione (GSSG/GSH), oxidized glutathione

(GSSG), and reduced glutathione (GSH) were significantly higher in the tumor than the surrounding muscle tissue, suggesting increased oxidative stress and the capacity to defend against oxidative stress (Fig. 1h). Beta-hydroxybutyrate/acetoacetate ratio (Fig. 1i), an indicator of mitochondrial NADH/NAD$^+$ ratio, was higher in sarcomas, indicating that during oxidative phosphorylation in mitochondria, the coupling of electron transport to ATP production was more efficient in muscle than in sarcomas. To validate the LC/MS findings, we independently measured NAD$^+$/NADH and NADP$^+$/NADPH levels in adjacent normal muscle and sarcomas using colorimetric assay kits. We found a significantly higher level of NAD$^+$/NADH in sarcomas compared to adjacent normal muscle (Supplementary Fig. 1c), suggesting increased glycolysis and proliferation in sarcomas. Similarly, higher levels of NADP + /NADPH were found in sarcomas, highlighting a potential increased glucose oxidation to the pentose phosphate pathway for nucleotide synthesis (Supplementary Fig. 1d). However, it is important to note that the BHB/AcAc ratio (Fig. 1i) is not necessarily coupled with cytosolic NADH/NAD$^+$, as mitochondrial and cytosolic NADH/ NAD$^+$ are separately regulated[25]. The colorimetric kit measures the whole cell NADH/ NAD$^+$, making it challenging to directly compare the results between the BHB/AcAc ratio and those obtained from the colorimetric kit. Regardless, these data indicate that in STS, precursors for macromolecule synthesis and glutathione defense systems were elevated, accompanied by a higher glutaminolysis. The metabolic difference was consistent with the metabolic requirement to support proliferating sarcoma cells and energy demand to support skeletal muscle contractions and other functions.

### $^{13}$C-Glucose infusion revealed different glucose-derived carbon utilization in sarcomas versus healthy muscle

Even though metabolite analysis revealed differences in the metabolic profile of sarcomas and adjacent muscles, it provided incomplete information to mechanistically understand the causes of the metabolic difference observed in Fig. 1. For instance, the increased glutaminolysis and the higher amounts of other non-essential amino acids in sarcomas compared to normal muscles suggest an increase in either de novo synthesis or uptake of exogenous NEAAs from plasma. To tackle these questions, we utilized a stable isotope tracing approach. We first performed euglycemic glucose infusions of [U-$^{13}$C]glucose in wildtype C57BL/6 mice and collected blood plasma at 0, 45 min, 1, 1.5, 2, 2.5, and 3 h post-infusion (Supplementary Fig. 2a). Consistent with a euglycemic infusion, plasma glucose and insulin levels were virtually unchanged throughout the infusion (Supplementary Fig. 2b, c). Furthermore, we used LC/MS to quantify [U-$^{13}$C]glucose enrichment in plasma and found that glucose steadily increased for the first 1.5 h of the infusion and remained stable for the remainder of the infusion (Supplementary Fig. 2d). In addition, we showed that glucose-derived carbon labeling into alanine, serine, proline, glycine, asparagine, glutamate, and glutamine reached a steady state in plasma at ~ 1.5 h post-infusion initiation (Supplementary Fig. 2e). These data demonstrate that [U-$^{13}$C] glucose reached a steady-state level by 3 h post-infusion initiation and labeled a number of NEAAs in plasma.

Because tumor metabolism is heavily influenced by the tumor microenvironment and relevant genetic lesions[21,26], we next investigated how P7NP sarcomas utilize different fuel sources to support increased energy and biosynthesis demand. Therefore, we first infused [U-$^{13}$C]glucose into P7NP sarcoma-bearing mice. At the end of the infusions, tumor, adjacent muscle, and blood plasma were collected and subjected to LC/MS (Fig. 2a). First, we measured total glucose in plasma (Fig. 2b) and then measured flux into glycolysis, TCA cycle, amino acid biosynthesis, and nucleotide biosynthesis (Fig. 2c). All values were normalized to plasma $^{13}$C-glucose enrichment to account for differences in plasma labeling. Consistent with enhanced glucose use in glycolysis, sarcomas had increased labeling of glycolytic metabolites and intermediates of nucleotide and amino acids biosynthesis. For instance, we found significantly higher glucose-derived carbon labeling into (1) glucose-6-phosphate oxidative pentose phosphate pathway intermediates, such as ribose-5-phosphate (Fig. 2d, e),

(2) bis-phosphoglycerate (Fig. 2f) and amino acids, such as serine and glycine (Fig. 2g, h), and (3) pyruvate and amino acids, such as alanine (Fig. 2i, j). Because recent data have revealed that glucose oxidation is elevated in some tumor types[27], we analyzed glucose utilization in the TCA cycle. Glucose provides carbon to the TCA cycle via two entry points. The first pathway is via the enzyme pyruvate dehydrogenase, which decarboxylates pyruvate to acetyl-CoA and can be assayed via the labeling pattern of $M + 2$ and $M + 4$ citrate and downstream metabolites, and the second

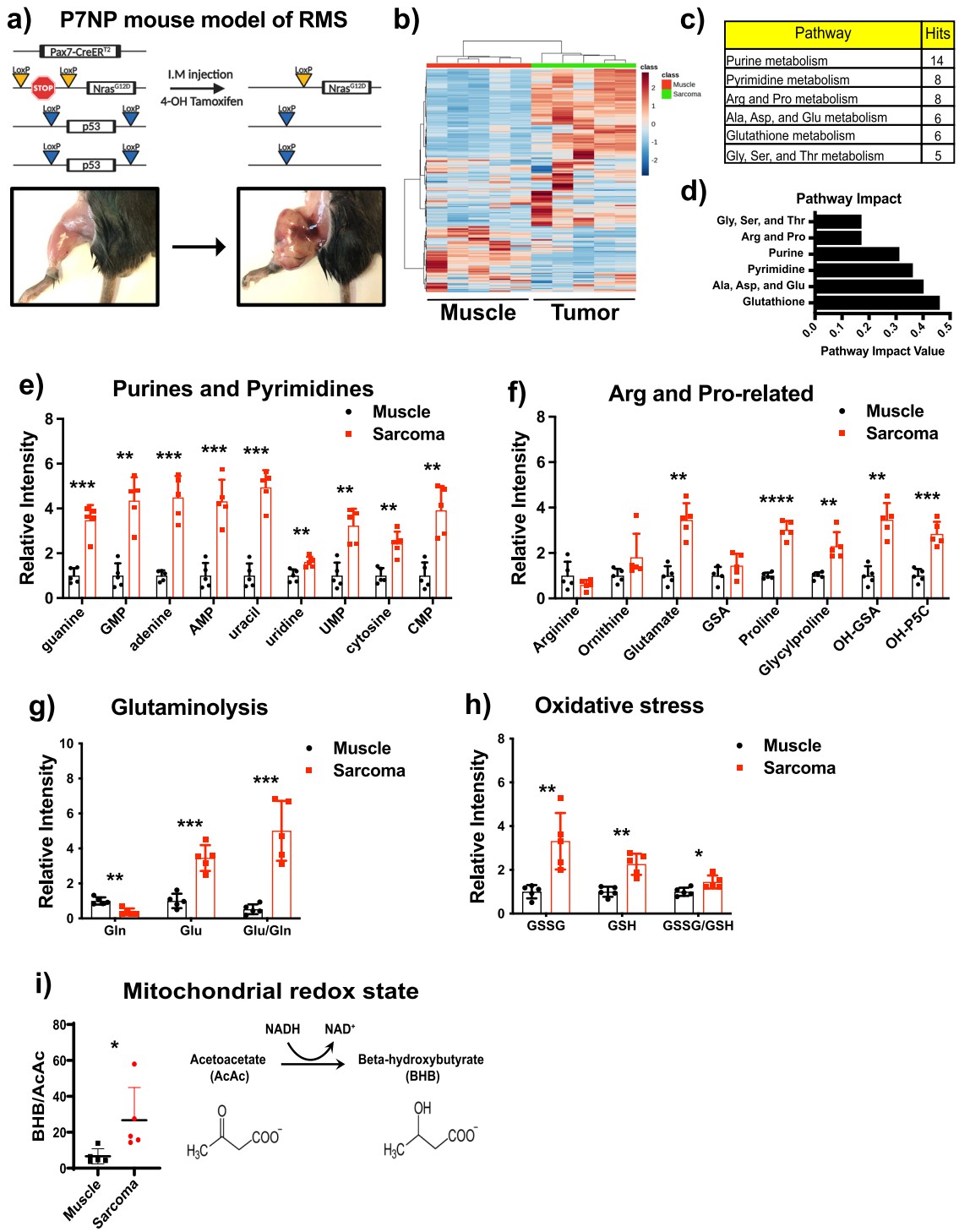

**Fig. 1 | Disruption of a broad range of metabolites in autochthonous soft-tissue sarcomas. a** Diagram of $Pax7^{CreERT2/+}$, $Nras^{LSL-G12D/+}$, $p53^{FL/FL}$ (P7NP) mouse model of soft tissue sarcoma generated by intramuscular (I.M.) injection of 4-hydroxytamoxifen. **b** Heatmap quantified metabolite abundance in primary soft tissue sarcomas ($n = 5$) and surrounding healthy muscles ($n = 5$) with LC/MS. **c** Pathway analysis of differentially regulated metabolites in healthy muscle and soft tissue sarcoma. **d** Pathway impact of differentially regulated metabolites in muscle and soft tissue sarcoma. Scatter bar graphs showed metabolite abundance in representative pathways, such as (**e**) Purine and pyrimidine biosynthesis, (**f**)

Arginine and proline metabolism, (**g**) Glutaminolysis, and (**h**) Oxidative stress. **i** Ratio of beta-hydroxybutyrate (BHB) to acetoacetate (AcAc) in muscle and sarcomas. Relative metabolite values were calculated by dividing the MS intensity of each metabolite by the corresponding MS intensity in the muscle. $p$-values in scatter bar graphs were calculated using multiple $t$-tests. All data were presented as mean ± S.D. $n$ number of mice; *$p < 0.05$, **$p < 0.01$, ***$p < 0.001$, ****$p < 0.0001$; RMS rhabdomyosarcoma, Arg Arginine, Pro Proline, Gln Glutamine, Glu Glutamate, GSH Reduced glutathione, GSSG Oxidized glutathione.

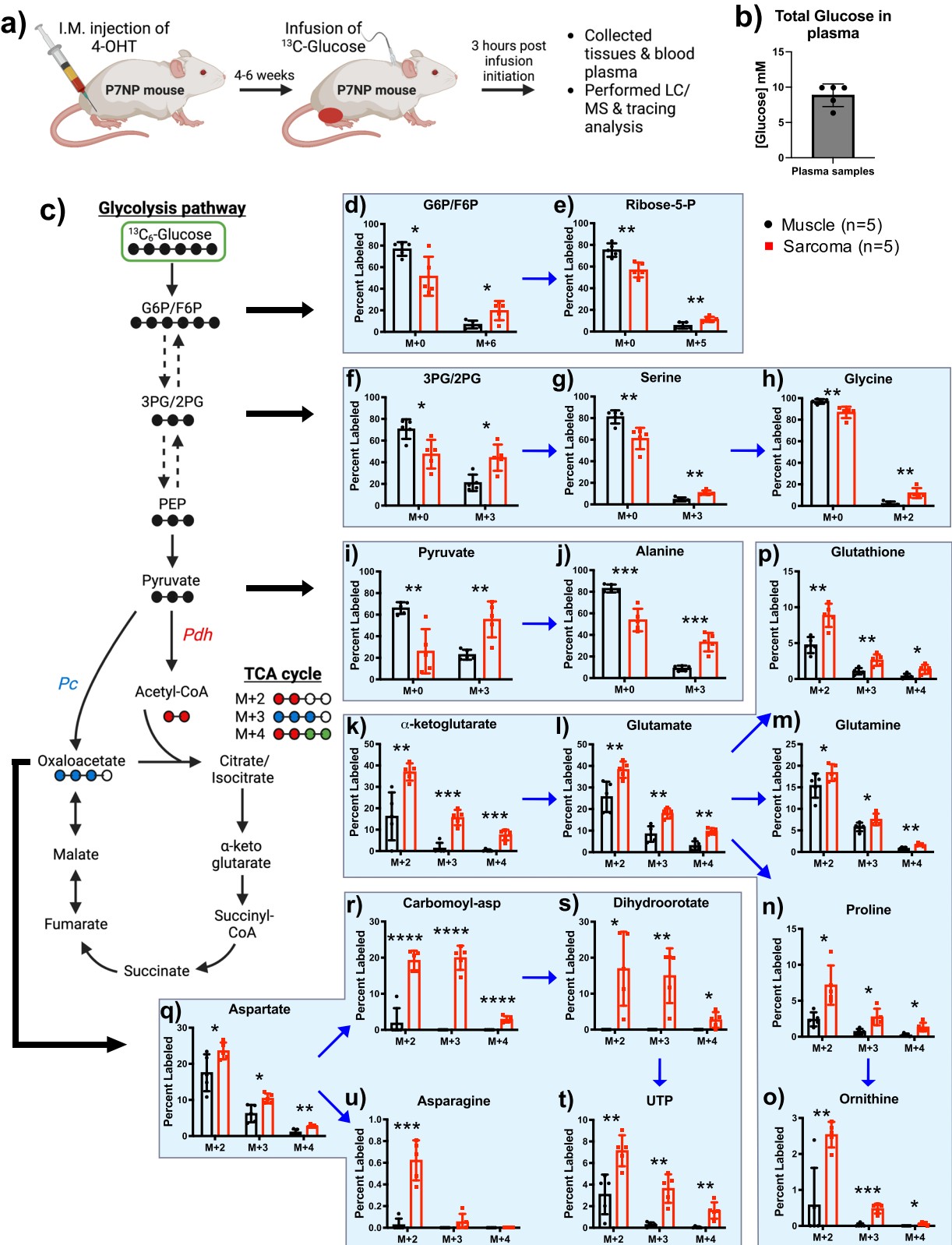

pathway is via the enzyme pyruvate carboxylase, which carboxylates pyruvate to oxaloacetate and can be assayed via the labeling pattern of M + 3 oxaloacetate and downstream metabolites. The percentage of unlabeled metabolites (M + 0) between sarcoma and normal muscle were shown in Supplementary Fig. 3a, b. In sarcomas compared to normal muscles, a significantly higher glucose-derived carbon labeling was found into α-

ketoglutarate (Fig. 2k) and amino acids biosynthesis substrates, such as glutamate, glutamine, proline, and Ornithine (Fig. 2l–o). Glucose-derived carbon labeling into glutathione was also significantly higher in sarcomas compared to normal muscles, suggesting an increased capacity to defend against oxidative stress (Fig. 2p). Similarly, in sarcomas compared to normal muscles, significantly higher labeling of glucose-derived carbon into

**Fig. 2 | ¹³C-Glucose infusion revealed different glucose-derived carbon utilization in sarcomas versus healthy muscles. a** Schematic representation of tumor development, ¹³C-glucose infusions, and LC/MS analysis of tumor, muscle, and plasma samples. **b** Total glucose in blood plasma at the time of sacrifice. **c** Schematic representation of how glucose-derived carbon flux through different catabolic and anabolic metabolites of glycolysis and TCA cycle. **d–u** ¹³C-glucose infusion showed labeling of downstream glucose metabolism into (**d**) Glucose-6-phosphate/fructose-6-phosphate (G6P/F6P), (**e**) Ribose-5-phosphate (Ribose-5-P), (**f**) 3-phosphoglycerate/2-phosphoglycerate (3PG/2PG), (**g**) Serine, (**h**) Glycine, (**i**) Pyruvate, (**j**)

Alanine, (**k**) α-ketoglutarate, (**l**) Glutamate, (**m**) Glutamine, (**n**) Proline, (**o**) Ornithine, (**p**) Glutathione, (**q**) Aspartate, (**r**) Carbomoyl-aspartate (Carbomoyl-asp), (**s**) Dihydroorotate, (**t**) Uridine-triphosphate (UTP), and (**u**) Asparagine. All tissue enrichments were normalized to the ¹³C-glucose enrichment in plasma. *p*-values in scatter bar graphs were calculated using multiple *t*-tests. All data were presented as means ± S.D. *n* number of mice; *$p < 0.05$, **$p < 0.01$, ***$p < 0.001$, ****$p < 0.0001$; 4-OHT 4-hydroxy tamoxifen, LC/MS Liquid chromatography/Mass spectrometry, Pdh Pyruvate dehydrogenase, Pc Pyruvate carboxylase.

aspartate (Fig. 2q) and downstream nucleotide biosynthesis metabolites, such as carbamoyl-aspartate, dihydroorotate, UTP, and asparagine (Fig. 2r–u). The data suggest that glucose-derived carbon labeling into glycolytic and TCA cycle metabolites was significantly elevated in sarcomas compared to normal muscles.

## Glutamine-derived carbon contributed to the TCA cycle and gluconeogenesis in sarcomas in vivo

To confirm glutamine steady-state level in plasma and glutamine-derived carbon incorporation in the NEAA pool, we also performed continuous infusions of [U-¹³C]glutamine in wildtype C57BL/6 mice and collected blood plasma at similar time points described in the glucose infusion experiment. The total amount of glutamine rose initially during the infusion but remained within the normal physiological range, i.e., < 1 mM (Supplementary Fig. 2f). Similarly, the absolute glucose level was reduced initially during the infusion but reached a steady state after 1 h (Supplementary Fig. 2g). Consistent with the glucose infusion experiment, our results showed that [U-¹³C]glutamine enrichment increased for the first 1.5 h and then stabilized for the remainder of the infusion (Supplementary Fig. 2h). We were able to detect [U-¹³C]glutamine labeling of alanine, serine, proline, glutamate, and arginine in the plasma, and in each case, the labeling reached an isotopic steady state within the first 1.5 h of the infusion (Supplementary Fig. 2i).

Although glutamine utilization by cancer cells during in vitro tissue culture is well described, the use of glutamine by tumors in vivo is less well understood, especially in autochthonous or spontaneous tumors. To determine whether and how P7NP sarcomas use glutamine to fuel their growth, we infused [U-¹³C]glutamine for 3 h and collected tumor and adjacent muscle tissue for LC/MS analyses to determine flux into gluconeogenesis, TCA cycle, amino acid biosynthesis, and nucleotide biosynthesis (Fig. 3a, b). All metabolite enrichments were normalized to plasma ¹³C-glutamine enrichment to eliminate differences in plasma labeling. The percentage of unlabeled metabolites (M + 0) between sarcoma and normal muscle were shown in Supplementary Fig. 3c–e. Labeled glutamine was delivered to the muscle and tumor tissue to a similar degree, resulting in a similar enrichment in tissue glutamine M + 5 (Fig. 3m). Consistent with enhanced glutamine catabolism enrichment, M + 3 and M + 5 glutamate (Fig. 3l) and α-ketoglutarate (Fig. 3k) were significantly higher in sarcomas relative to muscle. Furthermore, elevated levels of glutamine-derived carbon in glutathione (Fig. 3p), proline (Fig. 3n), and ornithine (Fig. 3o) were found in sarcomas compared to muscles. We also found that glutamine-supplied carbon for nucleotide biosynthesis via the TCA intermediates. In sarcomas, significant deposition of glutamine-derived carbon was observed in the de novo pyrimidine synthesis pathway with enhanced labeling in aspartate, carbamoyl-aspartate, dihydroorotate, and UTP (Fig. 3q–t). To explore the use of glutamine in other metabolic pathways, we looked at glutamine tracing into glycolytic intermediates M + 2 and M + 3 glucose-6-phosphate/fructose-6-phosphate, 2/3-phosphoglycerate, and pyruvate. Consistent with enhanced glutamine-derived gluconeogenesis, M + 2 and M + 3 glucose-6-phosphate/fructose-6-phosphate (Fig. 3d) and pyruvate (Fig. 3i) were elevated in sarcomas compared to muscle, respectively. However, we found similar labeling of glutamine-derived carbon into 2/3-phosphoglycerate (Fig. 3f) and glucose (Fig. 3c) in sarcomas and muscles. These data demonstrate that glutamine partially contributes to gluconeogenesis in vivo in autochthonous sarcomas. We also observed evidence that glutamine-derived carbon can be used in the oxidative pentose phosphate

pathway to synthesize ribose-5-phosphate (Fig. 3e). In addition to its use in nucleotide synthesis, glutamine can be used to synthesize several NEAAs. Indeed, glutamine use via gluconeogenesis results in the labeling of serine (Fig. 3g), glycine (Fig. 3h), and alanine (Fig. 3j) in primary sarcomas, suggesting that gluconeogenesis is an important feature of sarcoma metabolism that supports biomass accumulation. Increased glycolysis and glutaminolysis supply carbon into the TCA cycle and simultaneously increased gluconeogenesis, which draws carbon away from the complete oxidation, may appear counterproductive as carbon enters and exits the TCA cycle in opposite directions. However, this could offer growth advantages to tumor cells by shunting carbons away from complete oxo-phosphorylation and circulating carbons between glycolysis and TCA to favor biosynthesis pathways, which is consistent with the observations in Fig. 1. Taken together, these data demonstrate that in primary soft tissue sarcomas, glutamine catabolism is one of the major sources of carbon for the TCA cycle, which is subsequently used for gluconeogenesis and non-essential amino acid synthesis.

## Sarcomas switched from glucose to glutamine metabolism post-radiation therapy in vivo

Recent data suggest cancer cells rely on glutamine metabolism post-RT in vitro[28,29]. However, it is unclear if sarcoma cells also increase glutamine dependency post-RT in vivo, and if so, what is the fate of glutamine-derived carbon in different metabolite pathways in sarcoma cells? To understand the fate of glucose- and glutamine-derived carbon in sarcoma cells, we performed a stable infusion of [U-¹³C]glucose or [U-¹³C]glutamine in P7NP sarcoma-bearing mice 48 h after sham or 10 Gy RT, as shown in Fig. 4a. As mentioned previously, glucose-derived carbon enters the TCA cycle via acetyl-CoA or oxaloacetate, while glutamine-derived carbon enters the TCA cycle via α-ketoglutarate, as shown in Fig. 4b. Post-RT, cancer cells' survival depends on the efficient production of energy, removal of radiation-induced reactive oxygen species (ROS), and the synthesis of building blocks for cell division, such as nucleotides and amino acids. Glucose and glutamine both play an important role in glutathione synthesis, which helps eliminate ROS-induced DNA damage, and proline synthesis, which helps maintain protein synthesis and redox homeostasis in cancer cells. Therefore, we first look at glucose- and glutamine-derived carbon flux into glutathione and proline synthesis. We found that glutamine-derived carbon becomes a significant source for proline synthesis, but not for glutathione synthesis, compared to glucose-derived carbon post-RT (Fig. 4c), suggesting that glutamine-derived carbon might be diverted away from glutathione synthesis. Next, we looked at glucose- and glutamine-derived carbon flux into TCA cycle substrates to determine whether RT influences sarcoma carbon preference. We found that glutamine-derived carbon flux significantly increased in a majority of TCA cycle metabolites compared to glucose-derived carbon post-RT (Fig. 4d). In the absence of radiotherapy (- RT), the incorporation of glucose- and glutamine-derived carbon into the metabolites of glutamine utilization and TCA cycle pathways were similar in sarcomas (Fig. 4c, d). The data suggest that irradiation of P7NP sarcomas caused a shift from glucose to glutamine dependency for TCA cycle anaplerosis and proline synthesis.

## Glutaminase inhibition via pharmacological or genetic intervention sensitizes sarcomas to radiation therapy in vivo

Our data showed that P7NP sarcomas relied more on glutamine-derived carbon post-RT, so inhibition of glutaminolysis could potentially

**Article**

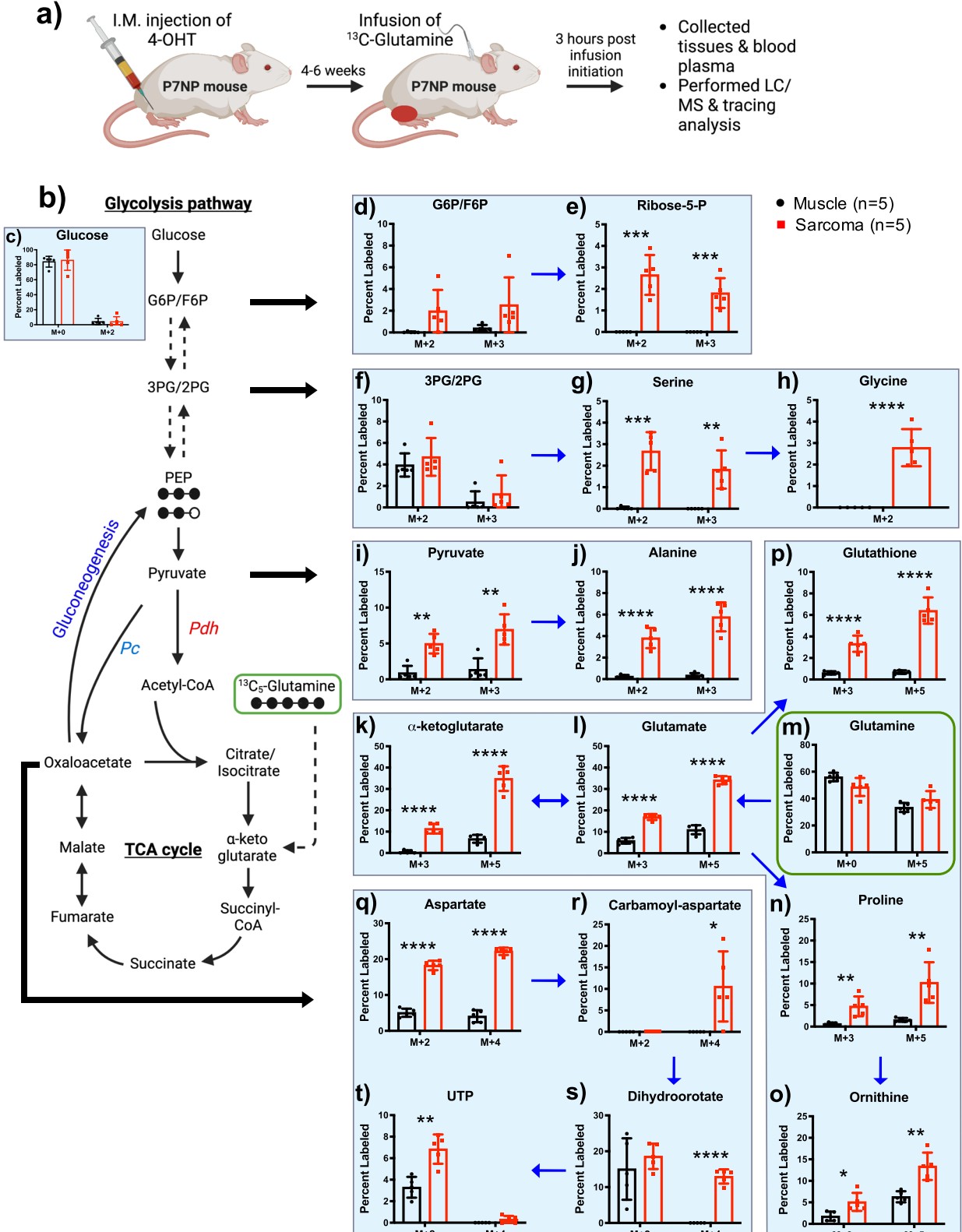

**Fig. 3 | Glutamine-derived carbon contributed to the TCA cycle and gluconeogenesis in sarcomas in vivo. a** Schematic representation of tumor development, ${}^{13}$C-glutamine infusions, and LC/MS analysis of tumor, muscle, and plasma samples. **b** Schematic representation of how glutamine-derived carbon flux through different catabolic and anabolic metabolites of glycolysis and TCA cycle. **c–t** ${}^{13}$C-glutamine infusion showed labeling of downstream glutamine metabolism into (**c**) Glucose, (**d**) Glucose-6-phosphate/fructose-6-phosphate (G6P/F6P), (**e**) Ribose-5-phosphate (Ribose-5-P), (**f**) 3-phosphoglycerate/2-phosphoglycerate (3PG/2PG), (**g**) Serine,

(**h**) Glycine, (**i**) Pyruvate, (**j**) Alanine, (**k**) α-ketoglutarate, (**l**) Glutamate, (**m**) Glutamine, (**n**) Proline, (**o**) Ornithine, (**p**) Glutathione, (**q**) Aspartate, (**r**) Carbamoyl-aspartate (carbamoyl-asp), (**s**) Dihydroorotate, and (**t**) Uridine-triphosphate (UTP). All tissue enrichments were normalized to the ${}^{13}$C-glutamine enrichment in plasma. *p*-values in scatter bar graphs were calculated using multiple *t*-tests. All data were presented as means ± S.D. *n* number of mice; *$p < 0.05$, **$p < 0.01$, ***$p < 0.001$, ****$p < 0.0001$; 4-OHT 4-hydroxy tamoxifen, LC/MS Liquid chromatography/ Mass spectrometry, Pdh Pyruvate dehydrogenase, Pc Pyruvate carboxylase.

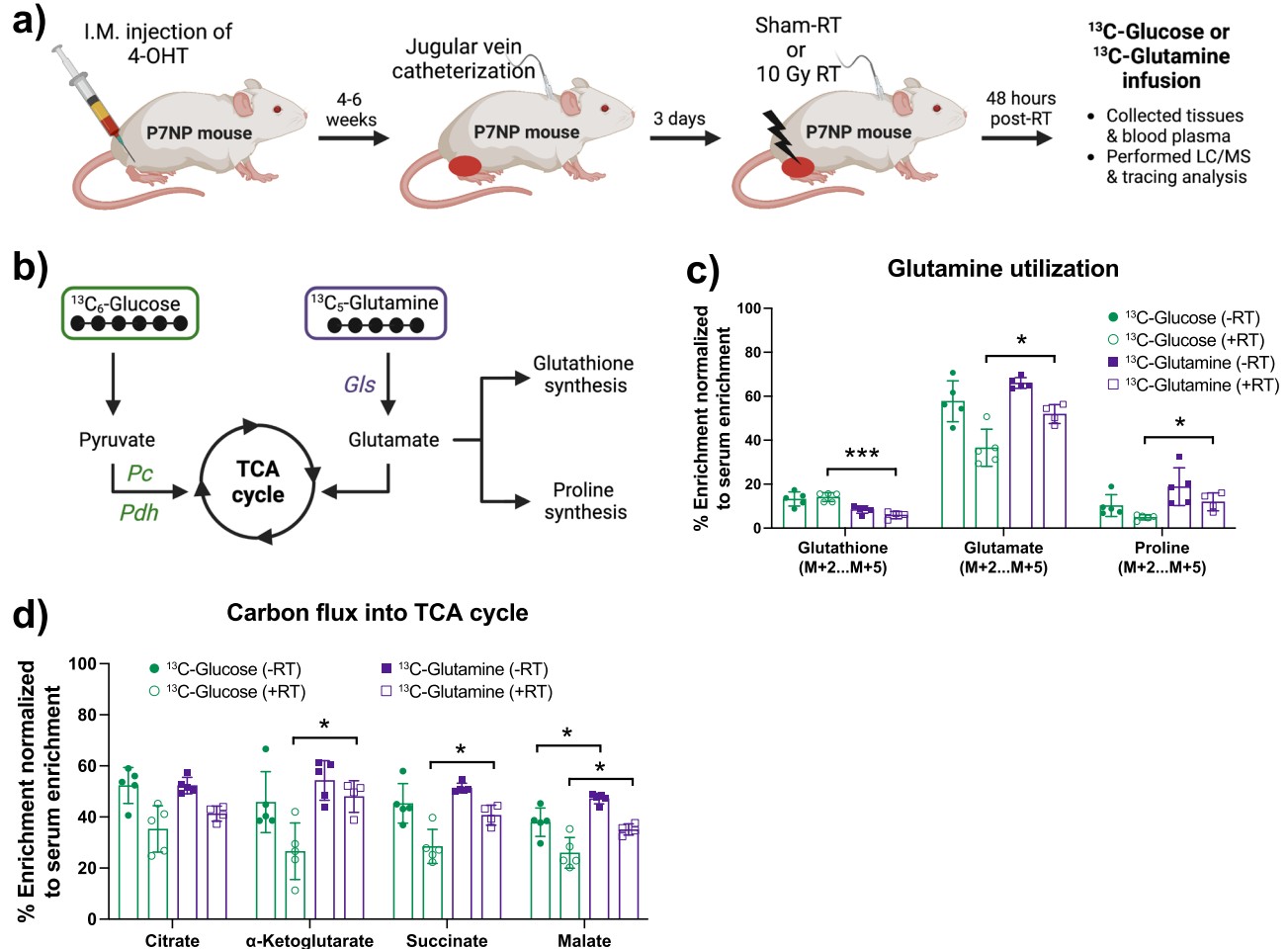

**Fig. 4 | Sarcomas switched from glucose to glutamine metabolism post-radiation therapy in vivo. a** Schematic representation of tumor development, tumor irradiation, $^{13}$C-glucose or $^{13}$C-glutamine infusions, and LC/MS analysis of tumor, muscle, and plasma samples. **b** Schematic representation of how glucose- or glutamine-derived carbon flux through different metabolites of TCA cycle and glutamine metabolism. **c** Scatter bar graph revealed glucose- and glutamine-derived carbon flux into glutathione and proline synthesis. **d** Scatter bar graph revealed

glucose- and glutamine-derived carbon flux into TCA cycle metabolites. All tissue enrichments were normalized to the $^{13}$C-glucose or $^{13}$C-glutamine enrichment in plasma. *p*-values in scatter bar graphs were calculated using multiple *t*-tests. All data were presented as means ± S.D. *$p < 0.05$, ***$p < 0.001$; 4-OHT 4-hydroxy tamoxifen, RT Radiation therapy, LC/MS Liquid chromatography/Mass spectrometry, Pdh Pyruvate dehydrogenase, Pc Pyruvate carboxylase, Gls1 Glutaminase.

radiosensitize sarcomas in vivo. The enzyme glutaminase (GLS1) plays a vital role in glutamine metabolism by generating glutamate, which is converted to α-ketoglutarate before entering the TCA cycle. To test the potential radiosensitization of sarcomas in vivo, we crossed P7NP mice with *Gls1*$^{fl/fl}$ mice to generate P7NP; *Gls1*$^{fl/fl}$ mice and induced primary sarcomas using an injection of 4-OHT into the gastrocnemius muscle (Fig. 5a). The deletion of *Gls1* in tumors led to an accumulation of glutamine which significantly reduced glutaminolysis in *Gls1*$^{fl/fl}$ compared to *Gls1*$^{+/+}$ and *Gls1*$^{fl/+}$ sarcomas (Fig. 5b). However, inhibition of glutaminolysis in *Gls1*$^{fl/fl}$ sarcomas did not impact time to tumor quintupling (Fig. 5c) and overall survival (Fig. 5d) compared to *Gls1*$^{+/+}$ or *Gls1*$^{fl/+}$ sarcomas, suggesting that deletion of glutaminase alone was insufficient to reduce sarcoma growth. Next, we irradiated *Gls1*-proficient (*Gls1*$^{fl/+}$) and *Gls1*-deficient (*Gls1*$^{fl/fl}$) sarcomas with or without 30 Gy (10 Gy/fraction for three consecutive days), as shown in Fig. 5e. We found that RT significantly prolonged the time to tumor quintupling of *Gls1*-deficient sarcomas (Fig. 5f), and thus significantly improved the overall survival of *Gls1*$^{fl/fl}$ mice compared to littermate control *Gls1*$^{fl/+}$ mice (Fig. 5g). Finally, we evaluated a pharmacological inhibitor of glutaminase (CB-839) to verify the radiosensitization of sarcomas through glutaminase inhibition. As shown in Fig. 5h, we irradiated *Gls1*$^{+/+}$ sarcomas with or without 30 Gy (10 Gy/fraction for three consecutive days) and

treated mice with vehicle or CB-839 twice a day till the end of the experiment via oral gavage. Similar to glutaminase deletion, we observed that glutaminase inhibition via CB-839 significantly delayed time to tumor quintupling (Fig. 5i) and increased overall survival (Fig. 5j) compared to vehicle-treated mice. Taken together, these data suggest that pharmacological targeting of glutaminase could improve sarcoma radiation response.

**Glutaminase deletion increased the innate immune response post-radiation therapy**

We performed label-free metabolomic and proteomic analyses to understand the possible mechanism(s) by which *Gls1* deletion radiosensitizes sarcomas. We irradiated *Gls1*$^{+/+}$ and *Gls1*$^{fl/fl}$ sarcomas with 10 Gy, sacrificed mice after 48 h, and collected tumors to evaluate molecular alterations that might be responsible for radiosensitization. First, we analyzed the unlabeled proteomic dataset and found that overall protein abundance was significantly reduced in *Gls1*$^{fl/fl}$ sarcomas post-RT compared to any other groups (Fig. 6a, b). We verified that glutaminase expression at the protein level was negligible in *Gls1*$^{fl/fl}$ sarcomas compared to *Gls1*$^{+/+}$ sarcomas, regardless of RT status (Fig. 6c). Next, we found that 490 and 217 proteins were differentially expressed in *Gls1*$^{fl/fl}$ sarcomas post 0 Gy and 10 Gy compared to *Gls1*$^{+/+}$ sarcomas, respectively (Supplementary Fig. 4).

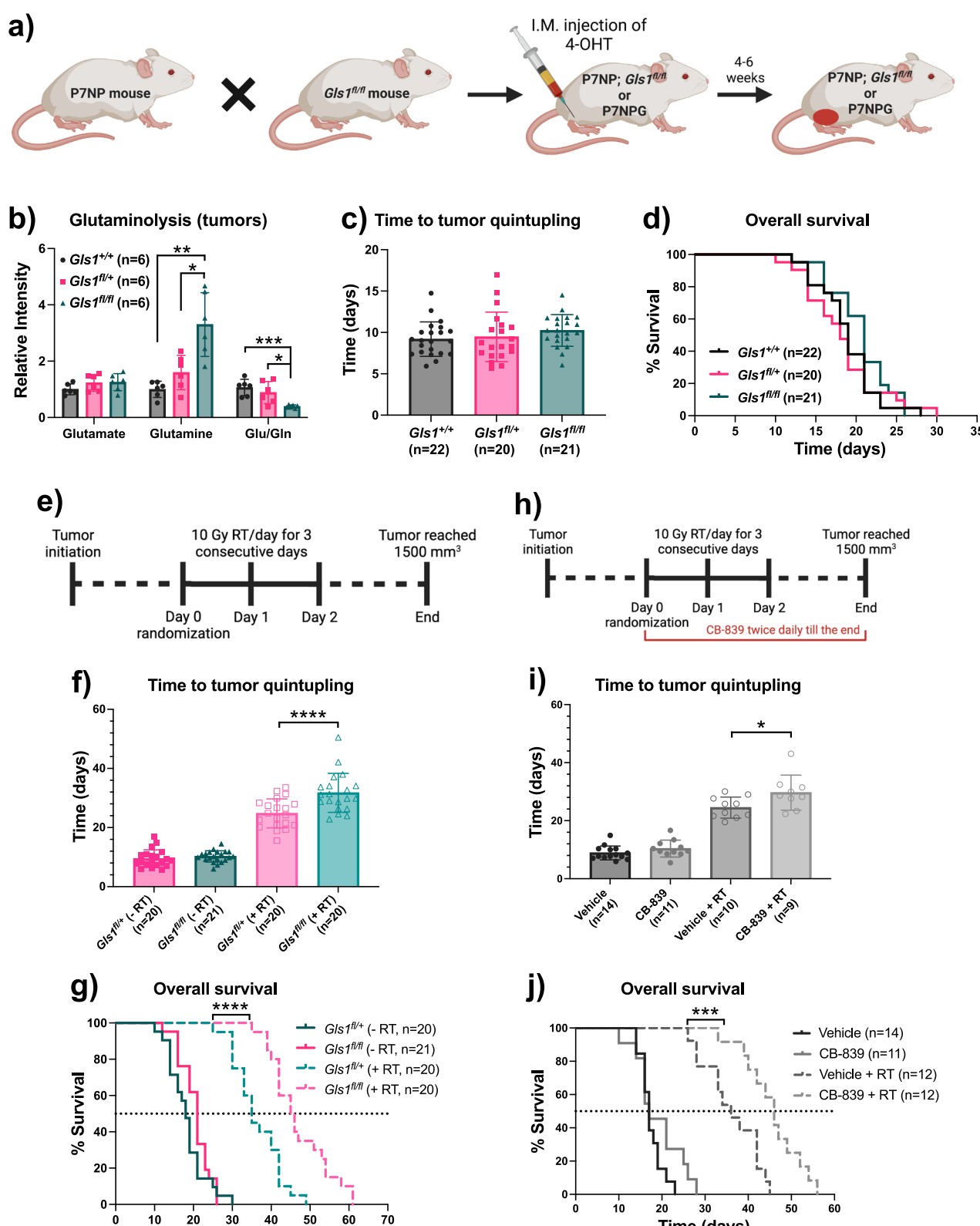

**Fig. 5 | Glutaminase inhibition via pharmacological or genetic interventions radiosensitized sarcomas in vivo.** **a** Schematic representation of *Gls1*-proficient (*Gls1*[+/+] and *Gls1*[fl/+]) and *Gls1*-deficient (*Gls1*[fl/fl]) P7NP sarcoma induction in mice. **b** Scatter bar graph showed the amount of glutaminolysis in *Gls1*[+/+], *Gls1*[fl/+] and *Gls1*[fl/fl] P7NP sarcomas. **c** Time to tumor quintupling, and (**d**) Overall survival of *Gls1*[+/+], *Gls1*[fl/+] and *Gls1*[fl/fl] P7NP sarcoma-bearing mice. **e** Schematic representation of experimental design evaluating RT response, (**f**) Time to tumor quintupling, and (**g**) Overall survival of *Gls1*[fl/+] and *Gls1*[fl/fl] P7NP sarcoma-bearing mice treated with

sham- or RT. **h** Schematic representation of experimental design evaluating glutaminase inhibitor (CB-839), (**i**) Time to tumor quintupling, and (**j**) Overall survival of *Gls1*[+/+] P7NP sarcoma-bearing mice treated with vehicle or CB-839 in combination with sham- or RT. *p*-values in scatter bar graphs were calculated using One-Way ANOVA. All data were presented as means ± S.D. *p*-values in Kaplan–Meier plots were calculated using Log-rank tests. *$p < 0.05$, **$p < 0.01$, ***$p < 0.001$, ****$p < 0.0001$; 4-OHT 4-hydroxy tamoxifen, RT Radiation therapy.

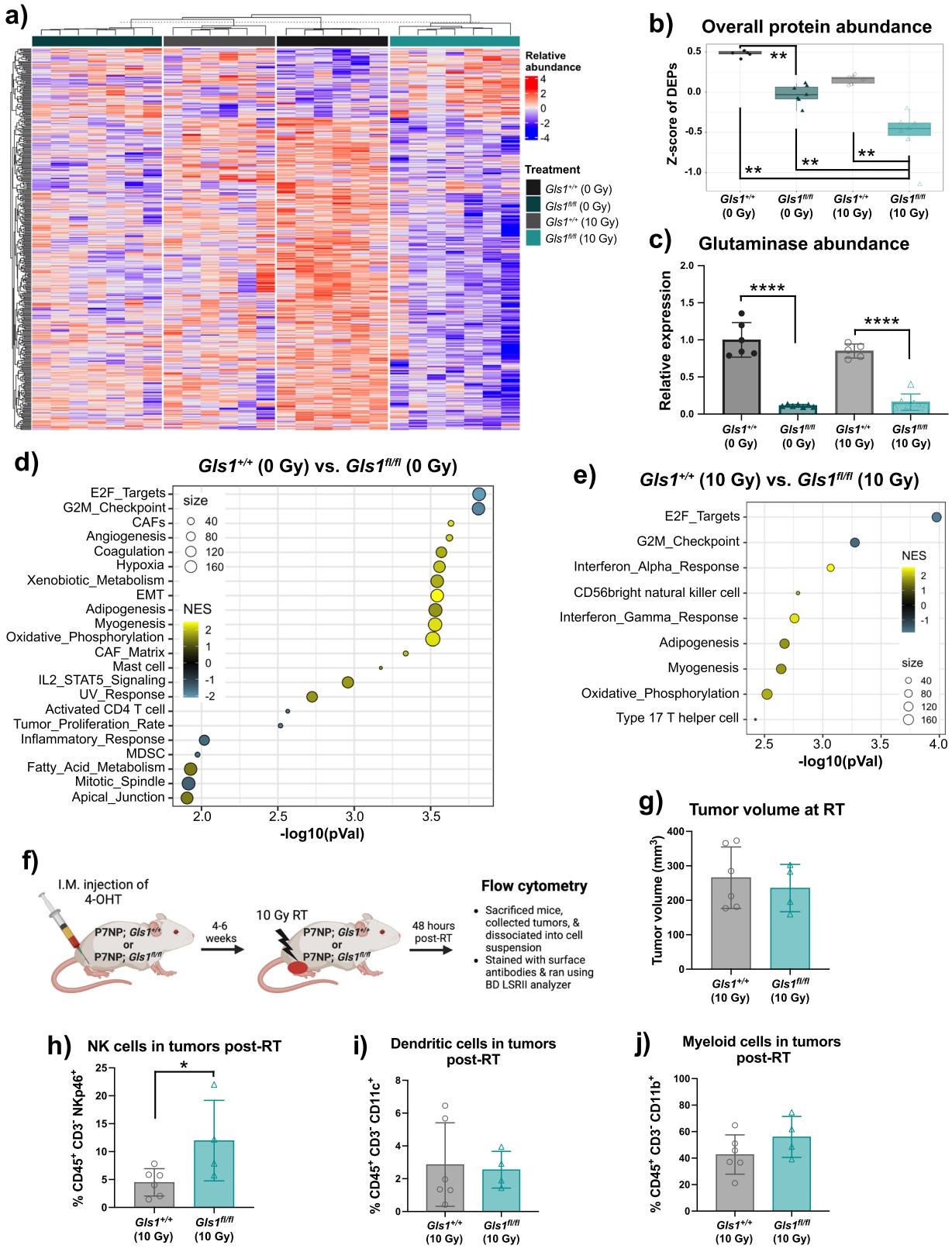

Hallmark pathway analysis revealed that *Gls1* deletion with or without RT decreased the expression of protein related to proliferation and translation (E2F targets and G2M checkpoint, Fig. 6d, e). Interestingly, *Gls1* deletion alone led to an immune suppressive tumor microenvironment facilitated by increased cancer-associated fibroblasts (CAF) & IL2_STAT5 signaling while simultaneously decreasing inflammatory & activated CD4 T cell

responses (Fig. 6d). In contrast, *Gls1* deletion post-RT increased innate immune response marked by elevated interferon-alpha, interferon-gamma, and natural killer (NK) cell responses (Fig. 6e). To validate these findings further, we performed flow cytometry to identify the presence of innate immune cells, such as NK, dendritic, and myeloid cells, in sarcomas post-RT. As shown in Fig. 6f, g, we irradiated an independent cohort of *Gls1*[+/+] or

**Fig. 6 | Glutaminase deletion increased innate immune response post-radiation therapy. a** Heatmap showed differentially expressed proteins identified using the analysis of variance (AOV) model across treatment groups. **b** Box and whiskers plot showed associated z-scores comparing the relative abundance of the differentially expressed proteins. **c** Scatter bar plot showed glutaminase abundance across treatment groups. **d** Gene set enrichment of proteins that were differentially expressed in *Gls1*[fl/+] (0 Gy) versus *Gls1*[fl/fl] (0 Gy) P7NP sarcomas. **e** Gene set enrichment of proteins that were differentially expressed in *Gls1*[fl/+] (10 Gy) versus *Gls1*[fl/fl] (10 Gy) P7NP sarcomas. **f** Schematic representation of flow cytometry experimental design. **g** Scatter bar plot showed tumor volume at the time of sarcoma irradiation. The flow

cytometry analysis revealed the presence of (**h**) Natural killer cells, (**i**) Dendritic cells, and (**j**) Myeloid cells in sarcoma 48 h post-RT. Each dot in the scatter bar plots represents an individual tumor sample. Associated *p*-values comparing *z*-scores across treatment groups were calculated using Wilcoxon test. *p*-values in scatter bar plots were calculated using multiple *t*-tests. All data were presented as means ± S.D. *p*-values in Kaplan–Meier plots were calculated using Log-rank tests. *$p < 0.05$, **$p < 0.01$, ***$p < 0.001$, ****$p < 0.0001$; RT Radiation therapy, CAFs Cancer-associated fibroblasts, EMT Epithelial-mesenchymal transition, MDSC Myeloid-derived suppressive cell, Gln Glutamine, Glu Glutamate.

*Gls1*[fl/fl] sarcomas with 10 Gy RT, sacrificed mice after 48 h to collect sarcomas, and dissociated them into single-cell suspension for antibody staining. Consistent with the results from proteomics analysis, we found significantly elevated levels of NK cells (Fig. 6h), but not dendritic (Fig. 6i) and myeloid cells (Fig. 6j), in *Gls1*[fl/fl] compared to *Gls1*[+/+] sarcomas post-RT. The elevated numbers of myeloid cells in primary mouse sarcomas after radiation therapy was expected, given our previously published work using a different genetically engineered and carcinogen-induced mouse sarcoma model[30].

Finally, we analyzed unlabeled polar and non-polar metabolites and noticed that *Gls1* deletion with or without RT caused subtle differences at the level of the metabolites (Supplementary Fig. 5a), but RT in *Gls1*[+/+] sarcomas significantly reduced overall metabolite abundance compared to *Gls1*[+/+] sarcomas (Supplementary Fig. 5d). In addition, we confirmed that *Gls1* deletion in sarcomas significantly reduced glutaminolysis compared to *Gls1* wildtype sarcomas, regardless of RT status (Supplementary Fig. 5b, c). Surprisingly, *Gls1* deletion with or without RT did not alter sarcomas' redox and oxidative state compared to *Gls1* wildtype sarcomas with or without RT, respectively (Supplementary Fig. 5e–h). Collectively, proteomic and flow cytometry datasets suggested that innate immune response is partly accountable for radiosensitizing *Gls1*-deficient sarcomas.

## Discussion

Understanding metabolic reprogramming in sarcomas may offer insights into tumor vulnerabilities and identify novel therapeutic targets. Recent studies have revealed that many factors, such as genetic lesions, metabolites within tumors, pH, oxygenation, and interactions with other cells in the tumor microenvironment (TME), including cancer-associated fibroblasts and immune cells, play a critical role in reprogramming the cancer cell metabolism[31]. However, many previous studies were performed either in vitro or in vivo using syngeneic transplant or immunodeficient xenograft tumor models, which exhibit limited inter- and intra-tumor heterogeneity and do not recapitulate gradual tumor development under immune surveillance. To overcome these limitations, we employed a genetically engineered mouse model of sarcoma that mimics embryonal rhabdomyosarcoma and performed both metabolomics and isotope tracing in vivo. Mechanistically, nutrient infusion of [U-$^{13}$C]-glucose and [U-$^{13}$C]-glutamine demonstrated that normal skeletal muscle uses glucose and glutamine primarily to meet energy demand, while sarcomas shunt glucose and glutamine carbons from oxidative metabolism into the biosynthesis of nucleotides, amino acids, and maintenance of redox status. Such reprogramming of metabolism allows sarcomas to maintain a relatively low energy status and optimizes carbon usage to promote proliferation. Our results demonstrated that glucose and glutamine are important carbon sources for STS in vivo and add to the list of metabolic reactions that could be targeted to limit sarcoma progression. Because of the relationship between the tumor microenvironment and nutrient requirements, it is essential to point out that skeletal muscle is quantitatively the most important site of glutamine synthesis[32,33]. Although the data presented here do not directly support metabolic crosstalk of muscle-sarcoma metabolism, these observations suggest a potential metabolic synergy between skeletal muscle glutamine production, which could fuel sarcoma growth. This notion should be explored in future studies. With the recent discovery that diet can impact tumor metabolism, development, and response to cancer

therapy[11], future research should also explore the impact of dietary interventions on muscle glutamine metabolism and concomitant sarcoma glutamine consumption and progression.

Interestingly, we discovered that sarcomas switched from glucose- to glutamine-derived carbon for proline synthesis and TCA cycle anaplerosis. Proline metabolism has emerged as a critical stress-responsive node in the metabolic network[34] and a critical mediator of the tumor growth[35]. Moreover, proline biosynthesis has been shown to be essential for mediating tumor growth in numerous tumor types, including melanoma, kidney cancer, prostate, and breast cancer[36–38]. In addition, dietary proline restriction reduced clear cell renal cell carcinoma proliferation and tumor xenograft growth, suggesting that proline availability is the predominant factor mediating tumor growth in these tumors[39]. Furthermore, our data indicate that glutamine-derived carbon is being used for proline over glutathione synthesis in sarcoma post-radiation therapy. Thus, inhibiting proline metabolism in combination with RT should be explored in future studies to further inhibit sarcoma growth in vivo.

Previous in vitro and in vivo studies highlighted how glutamine is critical in cancer and immune cell metabolism[40,41]. A recently published study has shown the importance of glutamine synthetase for nucleotide synthesis in pediatric sarcoma, mediating sarcoma cell proliferation[42]. Similarly, others showed that inhibiting glutamine metabolism delays undifferentiated pleomorphic sarcoma growth in vitro and in transplant mouse models in vivo[8]. Additional experiments have recently shown that inhibition of glutamine in tumors up-regulates oxidative metabolism in effector T cells leading to a long-lived and highly activated phenotype[43]. Here, we studied glutamine metabolism in an autochthonous sarcoma mouse model and showed that sarcomas switched from glucose- to glutamine-derived carbon for TCA cycle anaplerosis and proline synthesis. However, we found that inhibition of glutaminase alone was insufficient to impact Nras-driven primary sarcoma growth and overall survival of mice. This finding is interesting because glutaminase plays an essential role in a Myc-driven primary mouse model of hepatocellular carcinoma, where loss of one glutaminase allele significantly impacted tumor growth[44]. Taken together, these findings further support the notion that different oncogenic mutations and different tissues of origin influence nutrient dependence in cancer. Interestingly, we also revealed that inhibition of glutaminase radiosensitized autochthonous sarcomas in vivo, which might be mediated by the innate immune response. Glutaminase inhibitor CB-839 is currently being evaluated in the clinic for a combination treatment approach in non-small cell lung cancer patients. Thus, pharmacological inhibition of glutamine metabolism could be a viable option for STS patients undergoing standard-of-care radiotherapy. Finally, this study highlights the value of interrogating metabolic pathways in vivo settings by combining metabolomics, in vivo tracing approaches, and the use of genetically engineered mouse models of sarcomas. Results from this study define glutamine metabolism as a new metabolic pathway that could be targeted in combination with RT in STS.

## Methods
### Animal models
We have complied with all relevant ethical regulations for animal use. All animal procedures were approved by the Institutional Animal Care and Use

Committee (IACUC) at Duke University. Mouse models of soft-tissue sarcoma were generated on a mixed background (129/SvJae and C57BL/6) using a combination of alleles that have been previously described: Pax7[CreER-T2], p53[FL/FL], LSL-Nras[G12D] and ROSA26[mTmG] (P7NP)[24]. In addition, Gls1[fl/fl] (Gls1[tm2.1Isray]/J, Strain # 017894) mice were obtained from the Jackson Laboratory and crossed with P7NP mice to generate P7NP; Gls1[fl/fl] mice. For all experiments, male and female mice were utilized between the ages of six weeks and 6 months. All animals were kept in individually ventilated cages in a temperature- and humidity-controlled room.

### Primary mouse soft tissue sarcomas generation

The primary sarcomas were induced using an intramuscular (IM) injection of (Z)-4-hydroxytamoxifen (4-OHT) in the hind limb. 4-OHT was dissolved in 100% DMSO to achieve a 10 mg/ml concentration, and 50 μl of the solution was injected into the gastrocnemius muscle. For unlabeled metabolomics and proteomics analysis experiments, Gls1[+/+], Gls1[fl/+], and Gls1[fl/fl] P7NP sarcomas were allowed to reach 250–400 mm³ before sacrificing mice to collect tumor, adjacent normal muscle, and blood plasma. Similarly, ¹³C-labeled glucose or glutamine infusion experiments were initiated when Gls1[+/+] P7NP sarcomas reached 250–400 mm³. All tissue and blood plasma samples were snap-frozen using liquid nitrogen and stored at -80 °C for later use.

### In vivo ¹³C glucose and glutamine infusions

To perform in vivo nutrient infusions, chronic indwelling catheters were placed into the right jugular veins of mice, and animals were allowed to recover for 3 days before infusions. Jugular vein catheters, vascular access buttons, and infusion equipment were purchased from Instech Laboratories. Wildtype control and tumor-bearing mice were infused with [U-¹³C] glucose (Cambridge Isotope Laboratories, Cat # CLM-1396-1) for 3 h at a rate of 20 mg/kg/min (150 μL/hr). Similarly, [U-¹³C]glutamine (Cambridge Isotope Laboratories, Cat # CLM-1822-H-0.5) was infused for 3 h at a rate of 6 mg/kg/min (200 μL/hr) in wildtype control and tumor-bearing mice. For wildtype mice, blood was collected via the tail vein at 0, 30 min, 1, 1.5, 2, 2.5, and 3 h. Blood plasma was collected by centrifuging blood at 3,000 g for 15 min at 4 °C. At the end of infusions, tissues, and blood plasma samples were snap-frozen in liquid nitrogen and stored at -80 °C for further analyses.

### Metabolite extraction from tissue and plasma samples

The tumor sample was first homogenized in liquid nitrogen, and then 5–10 mg was weighed in a new Eppendorf tube. Next, 250 μl Ice cold extraction solvent (HPLC grade 80% methanol/water, v/v) was added to the tissue sample, and a pellet mixer was used to break down the tissue chunk further and form an even suspension, followed by the addition of 250 μl to rinse the pellet mixer. After incubation on ice for an additional 10 min, the tissue extract was centrifuged with the speed of 20,000 g at 4 °C for 10 min. 5 μl of the supernatant was saved in a -80 °C freezer until ready for further derivatization. The rest of the supernatant was transferred to a new Eppendorf tube and dried in a speed vacuum concentrator. The dry pellets were reconstituted into 30 μl (per 3 mg tissue) sample solvent (water:methanol:acetonitrile, 2:1:1, v/v), and 3 μl was injected into LC-HRMS. Similarly, metabolites were extracted from 20 μl of blood plasma using a similar protocol described for tissue samples.

### Metabolomic analysis of tissue and plasma samples

HPLC method—Ultimate 3000 UHPLC (Dionex) or Vanquish UHPLC (Thermo Fisher Scientific) was used for metabolite separation. Due to the instrumentation difference between Ultimate 3000 UHPLC and Vanquish UHPLC, different flow rates were used. For Ultimate 3000 UHPLC, the flow rate is: 0–5.5 min, 0.15 ml/min; 6.9–10.5 min, 0.17 ml/min; 10.6–17.9 min, 0.3 ml/min; 18–20 min, 0.15 ml/min. For Vanquish UHPLC, the flow rate is: 0–5.5 min, 0.11 ml/min; 6.9–10.5 min, 0.13 ml/min; 10.6–17.9 min, 0.25 ml/min; 18–20 min, 0.11 ml/min. For polar metabolite analysis, a hydrophilic interaction chromatography method (HILIC) with an Xbridge amide column (100 × 2.1 mm i.d., 3.5 μm; Waters) was used for compound

separation at room temperature. The mobile phase and gradient information were described previously[45].

Mass spectrometry—the Q Exactive Plus mass spectrometer or Orbitrap Exploris 480 mass spectrometer (Thermo Fisher Scientific) was equipped with a HESI probe. When Q Exactive Plus mass spectrometer was used, the relevant parameters were as listed: heater temperature, 120 °C; sheath gas, 30; auxiliary gas, 10; sweep gas, 3; spray voltage, 3.6 kV for positive mode and 2.5 kV for negative mode; capillary temperature, 320 °C; S-lens, 55. The resolution was set at 70,000 (at m/z 200). Maximum injection time (max IT) was set at 200 ms, and automated gain control (AGC) was set at $3 \times 10^6$. When Exploris 480 mass spectrometer was used, the relevant parameters were as listed: vaporizer temperature, 350 °C; ion transfer tube temperature, 300 °C; sheath gas, 35; auxiliary gas, 7; sweep gas, 1; spray voltage, 3.5 kV for positive mode and 2.5 kV for negative mode; RF-lens (%), 30. The resolution was set at 60,000 (at m/z 200). Automatic maximum injection time (max IT) and automated gain control (AGC) was used. A full scan range was set at 70 – 900 (m/z) with positive/negative switching when coupled with the HILIC method.

### NAD⁺/NADH and NADP⁺/NADPH measurement in sarcoma and normal muscle

To measure the redox and antioxidant status in normal muscle versus sarcoma, we induced primary tumors using injection of 4-OHT into the gastrocnemius muscle of P7NP mice. When the tumor reached a volume of 250–400 mm³, we euthanized the mice, collected the tumor and adjacent normal muscle, snap-froze them in liquid nitrogen, and stored them at -80 °C in a freezer. To measure the NAD⁺/NADH ratio, we took a small piece of tissue from each sample weighing 15–30 mg. We followed the steps described by the manufacturer in the NAD⁺/NADH Assay Kit (Abcam, Cat # ab65348) and measured the Total NADt (total NAD⁺ and NADH) & NADH using a colorimetric microplate reader at OD 450 nm. Similarly, to measure the NADP⁺/NADPH ratio, we took a small piece of tissue from each sample weighing 25–50 mg. We followed the steps described by the manufacturer in the NADP⁺/NADPH Assay Kit (Abcam, Cat # ab65349) and measured the Total NADPt (total NADP⁺ and NADPH) & NADPH using a colorimetric microplate reader at OD 450 nm. The ratios of NAD⁺/NADH and NADP⁺/NADPH in normal muscle and sarcoma were determined, and statistical differences were calculated using non-parametric t-tests.

### Proteomics analysis of sarcoma samples

Sample preparation—5% SDS in 50 mM triethylammonium bicarbonate (SDS/TEAB) was added to weighed samples at a 10:1 (v/w) followed by probe sonication. Additional denaturation was performed by heating at 80 °C for 5 min. After centrifugation, protein concentrations were determined by BCA. Next, 100 μg of each sample was adjusted to 50 μl with 5% SDS/TEAB. Samples were reduced by the addition of 5 μl of 110 mM DTT and heated at 80 °C for 15 min. After cooling, samples were alkylated by adding 75 μl of 250 mM iodoacetamide and incubated in the dark for 30 min. Finally, 6 μl of 12% phosphoric acid was added, followed by 400 μl of 90% MeOH/100 mM TEAB, and the samples were processed using S-traps microdevices. Samples were digested with 7 μg of Sequencing-Grade modified trypsin at 47 °C for 2 h. Eluted peptides were lyophilized and reconstituted in 30 μl of 1/2/97 (v/v/v) trifluoroacetic acid/acetonitrile/water followed by incubation in a sonicator bath and centrifugation. A study pool QC (SPQC) sample was prepared by mixing all samples in equal volumes (5 μl).

Quantitative data-independent acquisition (DIA) LC-MS/MS—samples were analyzed by DIA-microflow LC-MS/MS using an ACQUITY UPLC (Waters) interfaced to an Exploris 480 high-resolution tandem mass spectrometer (ThermoFisher Scientific). After direct injection, peptides (12.5 μl peptides/sample) were separated on a 1 mm × 10 mm 1.7 μm CSH C18 column (Waters) using a flow rate of 100 μl/min, a column temperature of 55 °C and a gradient using 0.1% (v/v) formic acid (FA) in H2O (mobile phase A) and 0.1% (v/v) formic acid (FA) in acetonitrile (MeCN, (mobile

phase B) @ 100 μl/min as follows: 0–60 min, 3–28% B; 60–60.5 min, 28–90% B; 60.5–62.5 min, 90% B; 62.5–63 min, 90-3% B; 63–67 min, 3% B. A tee was used post-column to introduce a solution of 50% (v/v) dimethyl sulfoxide/acetonitrile (DMSO/MeCN) at 6 μl per min. The LC was interfaced to the MS with an Optamax NG ion source under heated electrospray ionization (HESI) conditions with following tune parameters: sheath gas, 32; aux gas, 5; spray voltage, 3.5 kV; capillary temperature, 300 °C; aux gas heater temp, 125 °C. The DIA analysis used a staggered, overlapping window method[46] with a 60,000 resolution precursor ion (MS1) scan from 390–1020 m/z, AGC target of 1000% and maximum injection time (IT) of 60 ms and RF lens of 40%; data was collected in centroid mode. MS/MS was performed using tMS2 method with default charge state = 3,15,000 resolution, AGC target of 1000% and maximum IT of 22 ms, and a NCE of 30; data was collected in centroid mode. The DIA windows were generated using EncyclopeDIA with a mass range of 400–1000 and 77 × 16 m/z windows, with a 36-window cycle. The MS cycle time was 1.6 s, and the total injection-to-injection time was 67 min.

Quantitative analysis of DIA data—Raw MS data was demultiplexed and converted to \*.htrms format using HTRMS converter (v. 15.5.211111.50606; Biognosys) and processed in Spectronaut 15 (v. 15.5.211111.5060; Biognosys). A spectral library was built using Direct-DIA searches of all individual files. Searches used a Swissprot database with *mus musculus* taxonomy (downloaded on 03/05/21) and appended with contaminant sequences, namely human keratins and porcine trypsin (17,546 total entries). Search settings included trypsin specificity, up to 2 missed cleavages, fixed carbamidomethyl (Cys) and variable acetyl (protein-N-terminus), and oxidation (Met) modifications. Default extraction, calibration, identification, and protein inference settings were used for DIA analysis. Quantification was performed at the MS2 level using the MaxLFQ algorithm[47], $q$-value sparse settings (all precursors that passed a $q$-value included, with global imputation in runs where $q > 0.01$; no missing values), and cross-run normalization using $q$-value complete and median local normalization settings[48].

Protein set enrichment analysis—we used methods developed for gene set enrichment to calculate protein set enrichment. We calculated the differential abundance between each treatment group using a linear model. The beta-coefficients from the linear model were used as the ranks into Fast Gene Set Enrichment Analysis (*fgsea*, an R library). The gene sets used in this analysis were the Hallmark Gene Sets from mSigDB (https://www.gsea-msigdb.org/gsea/msigdb/), immune modules from Charoentong et al.[49], and additional tumor microenvironment modules from Bagev et al.[50].

## Irradiation of primary mouse sarcomas

To evaluate the impact of *Gls1* deletion, *Gls1*[fl/+] and *Gls1*[fl/fl] P7NP sarcomas were induced using 4-OHT injection into the gastrocnemius muscle. When the tumor reached 75–125 mm³, we randomized mice into sham- or RT (10 Gy/fraction for three consecutive days) treatment groups. RT was delivered using the X-RAD 225Cx micro-CT/micro-irradiator (Precision X-Ray), which can deliver image-guided radiation therapy with high precision to volumes as small as 1 mm³. Tumors were measured three times per week until the tumor volumes reached 1500 mm³. Tumor growth kinetics and survival of sarcoma-bearing mice were evaluated by plotting time to tumor quintupling (scatter bar plot) and overall survival (Kaplan–Meier plot). One-Way ANOVA and Log-rank Test were performed to calculate significance in scatter bar and Kaplan–Meier plots, respectively.

To evaluate the impact of the glutaminase inhibitor CB-839 (Calithera), *Gls1*[+/+] P7NP sarcomas were induced using 4 OHT injection into the gastrocnemius muscle. When the tumor reached 75–125 mm³, we randomized mice into a vehicle (25% (w/v) hydroxypropyl-β-cyclodextrin in 10 mmol/L citrate, pH 2) or CB-839 (200 mg/kg) treatment with or without RT (10 Gy/fraction for three consecutive days) treatment groups. RT was delivered using the X-RAD 225Cx micro-CT/micro-irradiator (Precision X-Ray), which can deliver image-guided radiation therapy with high precision to volumes as small as 1 mm³. Mice were treated with vehicle or CB-839 twice a day by gavage till the tumor volume reached 1500 mm³. Tumor

growth kinetics and survival of sarcoma-bearing mice were evaluated by plotting time to tumor quintupling (scatter bar plot) and overall survival (Kaplan–Meier plot). One-Way ANOVA and Log-rank Test were performed to calculate significance in scatter bar and Kaplan–Meier plots, respectively.

## Analyzing primary tumors by flow cytometry

To evaluate innate immune response in *Gls1*[+/+] and *Gls1*[fl/fl] P7NP sarcomas post-RT, we induced primary tumors using 4-OHT injection into the gastrocnemius muscle. When the tumor reached 150–400 mm³, we irradiated tumors with 10 Gy delivered using the Small Animal Radiation Research Platform or SARRP micro-CT/micro-irradiator (Xstrahl), which can deliver image-guided radiation therapy with high precision to volumes as small as 1 mm³. After 48 h, tumor-bearing mice were sacrificed, and tumors were collected for flow cytometry. Tumors were minced and digested using a tumor dissociation kit (Miltenyi Biotec, Cat # 130-096-730) for 45 min on a shaker at 37 °C. Dissociated tumor cells were strained through a 70 μm filter and washed with FACS buffer containing HBSS (Thermofisher, Cat # 14175095), 0.5 mM EDTA (Sigma-Aldrich, Cat # E7889-100 ML), and 2.5% FBS (Gibco, Cat # 16000044). Red blood cells were lysed using ACK lysis buffer (Lonza, Cat # 10-548E) and washed with FACS buffer. Finally, the total number of cells was counted using a hemocytometer, and 2.5 M cells/tumor were used for antibody staining.

Tumor cells were blocked with 1% purified rat anti-mouse CD16/CD32 (BD biosciences, Cat # 553142) for 15 min at room temperature before staining with cell surface antibodies. After blocking, tumor cells were stained with APC-Cy7 conjugated anti-mouse CD45 (BD biosciences, Cat # 557659), BV711 conjugated anti-mouse CD3 (BioLegend, Cat # 100241), PE conjugated anti-mouse NKp46 (BD biosciences, Cat # 560757), AF488 conjugated anti-mouse CD11c (BioLegend, Cat # 117313), and BV421 conjugated anti-mouse CD11b (BioLegend, Cat # 101235). Next, tumor cells were stained with Live/Dead Zombie Aqua fixable dye (BioLegend, Cat # 423101). All antibodies were used as per the company's recommendation. Finally, data were collected from ~400,000 cells/events with a BD LSRII cell analyzer and analyzed by FlowJo without the knowledge of the mouse genotype. During the data analysis, dead cells were first excluded using the viability dye Zombie Aqua, and then the single-cell population was isolated by plotting FSC-H vs. FSC-A. Immune cells were identified by plotting SSC-A vs. APC-Cy7-A (CD45 + ). After the identification of immune cells, two-parameter density plots were utilized to identify NK cells (NKp46-A vs. BV711-A), dendritic cells (AF488-A vs. BV711-A), and myeloid cells (BV412-A vs. BV711-A).

## Statistics and reproducibility

Because sex, age, and genetic drift could influence the treatment response[51,52], we used sex-balanced and age-matched littermate controls in all experiments to improve rigor and reproducibility. For statistical analysis, we performed a Student's $t$-test (two-tailed) to compare the means of the two groups. A one-way ANOVA and log-rank test were performed to calculate the significance of the scatter bar and Kaplan–Meier plots, respectively. In addition, proteomics data analysis was performed as mentioned in the method section labeled "Proteomics analysis of sarcoma samples." The exact number of animals utilized for each experiment was mentioned in the figure legends of primary and Supplementary Figures. Finally, we employed GraphPad Prism 10 software for statistical calculations and to generate panels shown in primary and Supplementary Figures.

## Reporting summary

Further information on research design is available in the Nature Portfolio Reporting Summary linked to this article.

## Data availability

The mass spectrometry proteomics data have been deposited to the ProteomeXchange Consortium via the PRIDE[53] partner repository with the dataset identifier PXD050936. The numerical source data behind the graphs

can be found in the Supplementary Data file. All other data are available from the corresponding author upon reasonable request.

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

## Acknowledgements

We thank Clay Rouse, DVM, for consultation in the jugular vein surgical procedure and members of the Kirsch lab and Locasale lab for helpful suggestions. We also thank Drs. Matthew Foster and Arthur Moseley at Duke Proteomics and Metabolomics Shared Resource for their help with the proteomics dataset. This work was supported by the National Institutes of Health R35CA197616 (D.G.K.) T32CA093240 (D.E.C.), Duke Cancer Institute Pilot Grant P30 CA014236 (D.G.K), RO1 CA193256 (J.W.L), a grant from the Slifka Foundation and Wendy Walk Foundation (D.G.K), AACR-QuadW Foundation Fellowship for Clinical/Translational Sarcoma Research 21-40-37-PATE (R.P.), and American Cancer Society RSG-16-214-01-TBE (J.W.L.). CB-839 was a gift from Calithera. In addition, we credit BioRender.com for schematics in the figures. Without using any premade templates available on the website, we created schematics such as mouse models and experimental design panels using the icons provided by BioRender.

## Author contributions

R.P. and D.E.C. performed all animal experiments, generated figures, and wrote the manuscript. K.T.K., N.T.W., and L.L. helped with animal experiments and maintained the mouse colony. A.A., L.D., and X.L. performed metabolomics analysis. X.L. and J.W.L. participated in experimental design and interpretation and contributed to manuscript revision. R.P., D.E.C., and D.G.K. designed all experiments, interpreted results, and wrote the manuscript. All authors provided input on the manuscript.

## Competing interests

D.G.K. is a co-founder of and stockholder in XRAD Therapeutics, which is developing radiosensitizers. D.G.K. is a member of the scientific advisory board and owns stock in Lumicell Inc, a company commercializing intraoperative imaging technology. None of these affiliations represents a conflict of interest with respect to the work described in this manuscript. D.G.K. is a coinventor on a patent for a handheld imaging device and is a coinventor on a patent for radiosensitizers. XRAD Therapeutics, Merck, Bristol Myers Squibb, and Varian Medical Systems provide research support to D.G.K., but this did not support the research described in this manuscript. J.W.L. advises Nanocare Technologies, Restoration Foodworks, and Cornerstone Pharmaceuticals, a company with interests in targeting cancer metabolism related to this work. The other authors have no conflicting interests.
