## [Peer Review File · Communications Biology]

Reviewers' comments:

Reviewer #1 (Remarks to the Author):

Author found a marked shift away from glucose consumption and towards glutamine metabolism after radiation therapy (RT) in vivo. This is a good work in sarcoma response to radiation therapy, which can help in resistance of sarcoma radiation therapy. Authors given a sufficient evidence for time selection for vivo animal experiments. The dependence of glutamine production after radiation is an old topic. It is a bright that the innate immunity after radiation therapy, which is not deeply described. so here are some suggestion.

1. Expand on the novelty of the article: While article touches on the changes in innate immunity, provide more in-depth analysis exploration on this aspect. This will enhance originality and significance of the study. For example, IHC of some immunity marker.
2. In line 78, the NADH/NAD⁺ should be measured by kit, and the NADPH/NADP⁺ should be measured for defense oxidative stress level.
3. The stable isotope tracing approach is an interesting results. so it should be demonstrate not only the percent labeled results, but also the total of abundance. in some key metabolites, the m0 should be showed.
4. the stable isotope tracing experiment, especially ¹³C5-glucose, it should be test the blood glucose.
5. In the ¹³C5-glutamine experiment, the G6P/F6P(m2,m3) is so high, the distinct values of each are how much. besides, the glucose also should be showed.

Reviewer #2 (Remarks to the Author):

The current report showed that glucose and glutamine are utilized more in sarcomas than in their adjacent healthy muscle tissues. Radiation therapy shifted glycolysis to glutamine metabolism in tumors, and deletion or inhibition of glutaminase resulted radiosensitization of sarcomas accompanied with increased expression of proteins involved in natural killer cell and interferon alpha/gamma responses.

The strength of this manuscript is the utilization of ample flux studies and mouse experiments to support the main conclusion. However, the key finding that the shift from glycolysis to glutamine in tumor occurs after radiation therapy was previously reported (despite in vitro approaches used) (Oxid Med Cell Longev, 2021. 2021: p. 5826932; Cell Rep, 2019. 28(5): p. 1136-1143.e4.). In addition, the underlying mechanisms of this switch as well as the altered immune responses were not explored. Thus, the current report is of limited novelty.

Points:

1. Fig. 1 shows that many metabolism pathways were alerted in sarcoma compared with normal tissue. Since glucose metabolism is also critical for tumor cell growth, it will be valuable if the authors could provide more information about the differences of glucose metabolism, including glycolysis and TCA cycle, between sarcoma and muscle tissue.

2. The authors showed that glutamine-derived carbon flux was increased in TCA cycle in tumor cells, compared to glucose-derived carbon post-RT. However, they did not show the percentage of contribution of these two carbon suppliers in tumor cells without radiation. Thus, more experiments are needed to support the statement “irradiation of P7NP sarcomas caused a shift from glucose to glutamine dependency”.
3. Numerous publications indicated that GLS expression and glutaminolysis are critical for tumor growth, but the authors showed that deletion of Gls1 or glutaminase inhibitor treatment did not affect sarcoma tumor growth and mice survival. The authors should explain the reason for this inconsistency.
4. The authors observed increased innate immune responses in Gls1-deleted mice post-RT, suggesting that glutaminolysis suppressed innate immunity. Will combined treatment of glycolysis inhibitor (e.g. 2-DG) with glutaminolysis inhibitor have an additive effect on the activation of innate immunity and suppression of tumor growth?
5. “Supplementary figure 2E” on page 5 should be “Supplementary figure 1E”.

Reviewer #3 (Remarks to the Author):

Overall, the manuscript presents a well-written and easy-to-follow investigation of the metabolism of rhabdomyosarcoma using a murine model that carries mutations frequently observed in human embryonal rhabdomyosarcoma. The study's findings are intriguing, particularly the shift from glucose to glutamine metabolism in sarcoma cells after radiation therapy (RT) and the potential radiosensitization observed upon Gls1 deletion or pharmacological inhibition of glutaminase.

Major Comments:

The authors should consider including a Gls1^{+/+} condition in Figures 5F-G. This addition would allow readers to compare and assess the differences more effectively between the Gls1-deficient and Gls1-normal sarcomas, providing a stronger basis for the conclusions drawn.

Figure 6, which describes the immune responses upon Gls1 deletion, appears somewhat general and does not seem to fully support the authors' conclusion regarding the increased innate immune response and changes in the sarcoma immune microenvironment. To bolster the conclusion, the authors should consider incorporating flow cytometry analysis data in Figure 6, specifically focusing on the markers used to identify the innate immune responses, and natural killer cell responses. This additional data will offer more robust evidence for the claims made.

In Figure 6, it is unclear how the authors specifically identify cancer-associated fibroblasts (CAFs) given that sarcoma cells share mesenchymal features with CAFs. It would be helpful if the authors could include markers or staining techniques used to identify CAFs in the figure, making it easier for readers to interpret the results accurately.

Reviewer #1 (Remarks to the Author) bolded in black text:

Author found a marked shift away from glucose consumption and towards glutamine metabolism after radiation therapy (RT) in vivo. This is a good work in sarcoma response to radiation therapy, which can help in resistance of sarcoma radiation therapy. Authors given a sufficient evidence for time selection for vivo animal experiments. The dependence of glutamine production after radiation is an old topic. It is a bright that the innate immunity after radiation therapy, which is not deeply described. so here are some suggestion.

We appreciate **Reviewer #1** identifying the importance of ^{13}C -labeled metabolite tracing and response to radiation therapy (RT) in a genetically engineered mouse model of sarcoma. As the reviewer requested, we performed additional experiments to bolster our manuscript. Please find our response to the Reviewers' comments in **red**, previous text already in the manuscript in **black**, and new data/text added to the manuscript in **purple**.

1. Expand on the novelty of the article: While article touches on the changes in innate immunity, provide more in-depth analysis exploration on this aspect. This will enhance originality and significance of the study. For example, IHC of some immunity marker.

We thank Reviewer #1 for making this excellent point. To further elaborate on the innate immune response, we performed flow cytometry analysis to quantify the presence of innate immune cells in sarcomas post-RT. Briefly, we induced *Gls1*-proficient and *Gls1*-deficient P7NP sarcomas in mice by injecting 4-hydroxytamoxifen into the gastrocnemius muscle. Next, we irradiated P7NP sarcomas with 10 Gy RT, collected them 48 hours later, and dissociated them into a single-cell suspension for fluorescent antibody staining. Flow cytometry analysis revealed that the presence of natural killer cells (NK), but not dendritic and myeloid cells, significantly increased in *Gls1*-deficient compared to *Gls1*-proficient P7NP sarcomas post-RT. These findings further validate the proteomics data shown in Figure 6. The detailed methods and results describing this experiment have been incorporated in the revised manuscript to strengthen the potential role of innate immunity in radiosensitizing *Gls1*-deficient sarcomas.

Analyzing primary tumors by flow cytometry

To evaluate innate immune response in *Gls1*^{+/+} and *Gls1*^{fl/fl} P7NP sarcomas post-RT, we induced primary tumors using 4-OHT injection into the gastrocnemius muscle. When the tumor reached 150-400 mm³, we irradiated tumors with 10 Gy delivered using the Small Animal Radiation Research Platform or SARRP micro-CT/micro-irradiator (Xstrahl), which can deliver image-guided radiation therapy with high precision to volumes as small as 1 mm³. After 48 hours, tumor-bearing mice were sacrificed, and tumors were collected for flow cytometry. Tumors were minced and digested using a tumor dissociation kit (Miltenyi Biotec, Cat # 130-096-730) for 45 minutes on a shaker at 37°C. Dissociated tumor cells were strained through a 70 μm filter and washed with FACS buffer containing HBSS (ThermoFisher, Cat # 14175095), 0.5 mM EDTA (Sigma-Aldrich, Cat # E7889-100ML), and 2.5% FBS (Gibco, Cat # 16000044). Red blood cells were lysed using ACK lysis buffer (Lonza, Cat # 10-548E) and washed with FACS buffer. Finally, the total number of cells was counted using a hemocytometer, and 2.5M cells/tumor were used for antibody staining.

Tumor cells were blocked with 1% purified rat anti-mouse CD16/CD32 (BD biosciences, Cat # 553142) for 15 minutes at room temperature before staining with cell surface antibodies. After blocking, tumor cells were stained with APC-Cy7 conjugated anti-mouse CD45 (BD biosciences, Cat # 557659), BV711 conjugated anti-mouse CD3 (BioLegend, Cat # 100241), PE conjugated anti-mouse NKp46 (BD biosciences, Cat # 560757), AF488 conjugated anti-mouse CD11c (BioLegend, Cat # 117313), and BV421 conjugated anti-mouse CD11b (BioLegend, Cat # 101235). Next, tumor cells were stained with Live/Dead Zombie Aqua fixable dye (BioLegend, Cat # 423101). All antibodies were used as per the company's recommendation. Finally, data were collected from ~400,000 cells/events with a BD LSRII cell analyzer and analyzed by FlowJo without the knowledge of the mouse genotype.

Glutaminase deletion increased the innate immune response post-radiation therapy

We performed label-free metabolomic and proteomic analyses to understand the possible mechanism(s) by which *Gls1* deletion radiosensitizes sarcomas. We irradiated *Gls1*^{+/+} and *Gls1*^{fl/fl} sarcomas with 10 Gy, sacrificed mice after 48 hours, and collected tumors to evaluate molecular alterations that might be responsible for radiosensitization. First, we analyzed the unlabeled proteomic dataset and found that overall protein abundance was significantly reduced in *Gls1*^{fl/fl} sarcomas post-RT compared to any other groups (Figure 6A, 6B). We verified that glutaminase expression at the protein level was negligible in *Gls1*^{fl/fl} sarcomas compared to *Gls1*^{+/+} sarcomas, regardless of RT status (Figure 6C). Next, we found that 490 and 217 proteins were differentially expressed in *Gls1*^{fl/fl} sarcomas post 0 Gy and 10 Gy compared to *Gls1*^{+/+} sarcomas, respectively (Supplementary figure 2). Hallmark pathway analysis revealed that *Gls1* deletion with or without RT decreased the expression of proteins related to proliferation and translation (E2F targets and G2M checkpoint, Figure 6D, 6E). Interestingly, *Gls1* deletion alone led to an immune suppressive tumor microenvironment characterized by increased cancer-associated fibroblasts (CAF) & IL2_STAT5 signaling while simultaneously decreased inflammatory & activated CD4 T cell responses (Figure 6D). Moreover, *Gls1* deletion post-RT showed an increased innate immune response marked by elevated interferon-alpha, interferon-gamma, and natural killer (NK) cell responses (Figure 6E). To validate these findings further, we performed flow cytometry to identify the presence of innate immune cells, such as NK, dendritic, and myeloid cells, in sarcomas post-RT. As shown in Figure 6F and 6G, we irradiated an

independent cohort of *Gls1*^{+/+} or *Gls1*^{fl/fl} sarcomas with 10 Gy RT, sacrificed mice after 48 hours to collect sarcomas, and dissociated them into single-cell suspension for antibody staining. Consistent with the results from proteomics analysis, we found significantly elevated levels of NK cells (Figure 6H), but not dendritic (Figure 6I) and myeloid cells (Figure 6J), in *Gls1*^{fl/fl} compared to *Gls1*^{+/+} sarcomas post-RT. The elevated numbers of myeloid cells in primary mouse sarcomas after radiation therapy was expected, given our previously published work using a different genetically engineered and carcinogen-induced mouse sarcoma model [XXXX].

Finally, we analyzed unlabeled polar and non-polar metabolites and noticed that *Gls1* deletion with or without RT caused subtle differences at the level of the metabolites (Supplementary figure 5A), but RT in *Gls1*^{+/+} sarcomas significantly reduced overall metabolite abundance compared to *Gls1*^{+/+} sarcomas

Figure 6:

(Supplementary figure 5D). In addition, we confirmed that *Gls1* deletion in sarcomas significantly reduced glutaminolysis compared to *Gls1* wildtype sarcomas, regardless of RT status (Supplementary figure 5B, 5C). Surprisingly, *Gls1* deletion with or without RT did not alter sarcomas' redox and oxidative state compared to *Gls1* wildtype sarcomas with or without RT, respectively (Supplementary figure 5E - 5H). Collectively, proteomic and flow cytometry datasets suggested that innate immune response is partly accountable for radiosensitizing *Gls1*-deficient sarcomas.

Figure 6: Glutaminase deletion increased innate immune response post-radiation therapy. (A) Heatmap showed differentially expressed proteins identified using the analysis of variance (AOV) model across treatment groups. (B) Box and whiskers plot showed associated z-scores comparing the relative abundance of the differentially expressed proteins. (C) Scatter bar plot showed glutaminase abundance across treatment groups. (D) Gene set enrichment of proteins that were differentially expressed in *Gls1^{fl/+}* (0 Gy) versus *Gls1^{fl/fl}* (0 Gy) P7NP sarcomas. (E) Gene set enrichment of proteins that were differentially expressed in *Gls1^{fl/+}* (10 Gy) versus *Gls1^{fl/fl}* (10 Gy) P7NP sarcomas. (F) Schematic representation of flow cytometry experimental design. (G) Scatter bar plot showed tumor volume at the time of sarcoma irradiation. The flow cytometry analysis revealed the presence of (H) Natural killer cells, (I) Dendritic cells, and (J) Myeloid cells in sarcoma 48 hours post-RT. Each dot in the scatter bar plots represents an individual tumor sample. Associated p-values comparing z-scores across treatment groups were calculated using Wilcoxon test. p-values in scatter bar plots were calculated using multiple t-tests. All data were presented as means \pm S.D. p-values in Kaplan-Meier plots were calculated using Log-rank tests. * $p < 0.05$, ** $p < 0.01$, *** $p < 0.001$, **** $p < 0.0001$; RT – Radiation therapy; CAFs – Cancer-associated fibroblasts; EMT – Epithelial-mesenchymal transition; MDSC – Myeloid-derived suppressive cell, Gln – Glutamine, Glu – Glutamate.

2. In line 78, the NADH/NAD⁺ should be measured by kit, and the NADPH/NADP⁺ should be measured for defense oxidative stress level.

As suggested by Reviewer #1, we performed an independent experiment in which sarcoma and adjacent normal muscle tissue samples were collected to detect NAD⁺/NADH and NADP⁺/NADPH using colorimetric assay kits. Briefly, the colorimetric analysis revealed significantly higher NAD⁺/NADH and relatively higher NADP⁺/NADPH in sarcomas compared to adjacent normal muscle samples. The higher NAD⁺/NADH ratio indicates a boost for aerobic glycolysis, while an elevated NADP⁺/NADPH ratio implies consistent anabolic demand for nucleotide synthesis. These results support the liquid chromatography/Mass spectrometry findings of sarcoma versus adjacent normal muscle shown in Figure 1. However, it's important to note that the BHB/AcAc ratio (Figure 1I) is not necessarily coupled with cytosolic NADH/NAD⁺, as mitochondrial and cytosolic NADH/ NAD⁺ are separately regulated [1]. The colorimetric kit measures the whole cell NADH/NAD⁺, making it challenging to directly compare the results between the BHB/AcAc ratio and those obtained from the colorimetric kit. We have highlighted this point in the revised manuscript.

NAD⁺/NADH and NADP⁺/NADPH measurement in sarcoma and normal muscle

To measure the redox and antioxidant status in normal muscle versus sarcoma, we induced primary tumors using injection of 4-OHT into the gastrocnemius muscle of P7NP mice. When the tumor reached a volume of 250-400 mm³, we euthanized the mice, collected the tumor and adjacent normal muscle, snap-froze them in liquid nitrogen, and stored them at -80°C in a freezer. To measure the NAD⁺/NADH ratio, we took a small piece of tissue from each sample weighing 15-30 mg. We followed the steps described by the manufacturer in the NAD⁺/NADH Assay Kit (Abcam, Cat # ab65348) and measured the Total NADt (total NAD⁺ and NADH) & NADH using a colorimetric microplate reader at OD 450 nm. Similarly, to measure the NADP⁺/NADPH ratio, we took a small piece of tissue from each sample weighing 25-50 mg. We followed the steps described by the manufacturer in the NADP⁺/NADPH Assay Kit (Abcam, Cat # ab65349) and measured the Total NADPt (total NADP⁺ and NADPH) & NADPH using a colorimetric microplate reader at OD 450 nm. The ratios of NAD⁺/NADH and NADP⁺/NADPH in normal muscle and sarcoma were determined, and statistical differences were calculated using non-parametric t-tests.

Disruption of a broad range of metabolites in autochthonous soft-tissue sarcomas

In order to better understand the metabolic adaptations that enable the outgrowth of soft-tissue sarcomas (STS), we utilized a genetically engineered mouse model that mimics rhabdomyosarcoma (RMS) that we previously characterized whereby intramuscular injection of 4-hydroxytamoxifen activated a muscle satellite cell-specific CreER^{T2} to turn on expression of an oncogenic *Nras*^{G12D} allele and delete both *p53* alleles [2], resulting in local tumor formation (P7NP sarcoma) within 4-6 weeks (Figure 1A). Once tumors reached ~250mm³, we collected the sarcomas and the surrounding healthy muscle tissues and performed liquid chromatography/mass spectrometry (LC/MS). A heatmap of the identified metabolites revealed significant differences between sarcoma and muscle tissue (Figure 1B). **The abundance of glycolysis and the TCA cycle metabolites were significantly higher in sarcomas compared to muscle, as shown in Supplementary figures 1A and 1B.** In addition, the top six pathways

Supplementary figure 1: Differentially regulated metabolites in normal muscle versus sarcoma. Scatter bar graphs showed metabolite abundance in (A) Glycolysis and (B) TCA cycle pathways. p-values were calculated using multiple t-tests. Comparison of the redox and antioxidant defense state of normal muscle versus sarcoma. The ratio of (C) NAD⁺/NADH highlighting catabolism and redox state, and (D) NADP⁺/NADPH highlighting anabolism and antioxidant defense state of normal muscle versus sarcoma. p-values were calculated using t-tests. All data are represented as means ± SD; *** p < 0.001; n = number of tissue samples.

that were different between tumor and muscle were related to nucleotide, amino acid, and glutathione metabolism (Figure 1C, 1D). Nucleotides serve as the building blocks for DNA and RNA synthesis, and the elevated nucleotide levels were consistent with the increased need for tumor cell proliferation (Figure 1E). Arginine and proline-related metabolites were also significantly higher in sarcomas compared to adjacent muscles (Figure 1F). Most importantly, glutaminolysis was significantly elevated in sarcomas compared to adjacent muscle, suggesting sarcomas' reliance on glutamine metabolism (Figure 1G). Further, the ratio of oxidized to reduced glutathione (GSSG/GSH), oxidized glutathione (GSSG), and reduced glutathione (GSH) were significantly higher in the tumor than the surrounding muscle tissue, suggesting increased oxidative stress and the capacity to defend against oxidative stress (Figure 1H). Beta-hydroxybutyrate/acetoacetate ratio (figure 1I), an indicator of mitochondrial NADH/NAD⁺ ratio, was higher in sarcomas, indicating that during oxidative phosphorylation in mitochondria, the coupling of electron transport to ATP production was more efficient in muscle than in sarcomas. **To validate the LC/MS findings, we independently measured NAD⁺/NADH and NADP⁺/NADPH levels in adjacent normal muscle and sarcomas using colorimetric assay kits. We found a significantly higher level of NAD⁺/NADH in sarcomas compared to adjacent normal muscle (Supplementary figure 1C), suggesting increased glycolysis and proliferation in sarcomas. Similarly, higher levels of NADP⁺/NADPH were found in sarcomas, highlighting a potential increased glucose oxidation to the pentose phosphate pathway for nucleotide synthesis (Supplementary figure 1D). However, it is important to note that the BHB/AcAc ratio (Figure 1I) is not necessarily coupled with cytosolic NADH/NAD⁺, as mitochondrial and cytosolic NADH/ NAD⁺ are separately regulated [1]. The colorimetric kit measures the whole cell NADH/ NAD⁺, making it challenging to directly compare the results between the BHB/AcAc ratio and those obtained from the colorimetric kit. Regardless, these data indicate that in STS, precursors for macromolecule synthesis and glutathione defense systems were elevated, accompanied by a higher glutaminolysis. The metabolic difference was consistent with the metabolic requirement to support proliferating sarcoma cells and energy demand to support skeletal muscle contractions and other functions.**

3. The stable isotope tracing approach is an interesting results. so it should be demonstrate not only the percent labeled results, but also the total of abundance.in some key metabolites, the m0 should be showed.

In response to Reviewer #1's suggestion, we have included a supplementary figure 3 in the manuscript that displays the M+0 or unlabeled metabolites in key metabolic pathways analyzed in ¹³C-Glucose and ¹³C-Glutamine tracing experiments that were conducted in Figures 2 and 3. Please see supplementary figure 3 below.

Supplementary figure 3: Unlabeled (M+0) metabolites abundance in different metabolic pathways. Percentage unlabeled (M+0) metabolites involved in (A) Glutamine utilization pathway, and (B) Nucleotide biosynthesis pathway during [U-¹³C]glucose infusion in P7NP sarcoma-bearing mice (n=5). Percentage unlabeled (M+0) metabolites involved in (C) Gluconeogenesis pathway, (D) Glutamine utilization pathway, and (E) Nucleotide biosynthesis pathway during [U-¹³C]glutamine infusion in P7NP sarcoma-bearing mice (n=5). p-values in scatter bar graphs were calculated using multiple t-tests. All data were presented as means ± S.D. * p < 0.05, ** p < 0.01, *** p < 0.001, **** p < 0.0001. n = number of mice.

4. the stable isotope tracing experiment, especially ¹³C5-glucose, it should be test the blood glucose.

Figure 2B - Total glucose in blood plasma at the time of sacrifice.

We appreciate the reviewer's comment on the potential impact of blood glucose for the *in vivo* tracing experiments. Indeed, before undertaking our *in vivo* ¹³C-Glucose and ¹³C-Glutamine tracing experiments in sarcoma-bearing mice, we performed an in-depth analysis of blood glucose during ¹³C-Glucose or ¹³C-Glutamine infusion to identify a window different time points with steady-state glucose levels, which would not impact the *in vivo* tracing experiments. As shown in supplementary figures 2B and 2G, blood glucose levels did not change significantly, with stabilization occurring after 90 minutes and remaining constant throughout the infusion. For every ¹³C-Glucose experiment, we also measured plasma glucose from blood collected at the time of sacrifice. Initially, we did not include the blood glucose levels in our results of the tracing experiment because it did not change significantly throughout the infusion.

However, based on Reviewer #1's comment, we now realize the importance of presenting the plasma glucose level during infusion. This information is now included in revised Figure 2B.

5. In the $^{13}\text{C}_5$ -glutamine experiment, the G6P/F6P(m2,m3) is so high, the distinct values of each are how much. besides, the glucose also should be showed.

We agree with the reviewer's comment that labeled G6P/F6P (Figure 3C) in the glutamine tracing experiment is significantly higher in sarcoma compared to normal muscle. We therefore revisited all datasets and noted that an error in the calculation of the percentage labeled G6P/F6P. We have updated the scatter bar plot in revised Figure 3D with the correct values. These data still show an abundance of G6P/F6P (M+2 and M+3) in sarcoma versus normal muscle below. In addition, as the reviewer suggested, the final end product of gluconeogenesis in the glutamine tracing experiment, glucose (Figure 3C), was incorporated in the revised Figure 3. We were unable to detect M+3 glucose in the glutamine tracing experiment, and hence, only M+0 and M+2 glucose is shown (Figure 3C). Please see the revised Figure 3 shown below.

Figure 3:

Reviewer #2 (Remarks to the Author):

The current report showed that glucose and glutamine are utilized more in sarcomas than in their adjacent healthy muscle tissues. Radiation therapy shifted glycolysis to glutamine metabolism in tumors, and deletion or inhibition of glutaminase resulted radiosensitization of sarcomas accompanied with increased expression of proteins involved in natural killer cell and interferon alpha/gamma responses.

The strength of this manuscript is the utilization of ample flux studies and mouse experiments to support the main conclusion. However, the key finding that the shift from glycolysis to glutamine in tumor occurs after radiation therapy was previously reported (despite *in vitro* approaches used) (Oxid Med Cell Longev, 2021. 2021: p. 5826932; Cell Rep, 2019. 28(5): p. 1136-1143.e4.). In addition, the underlying mechanisms of this switch as well as the altered immune responses were not explored. Thus, the current report is of limited novelty.

We appreciate Reviewer #2's identification of the strength of the manuscript as being extensive *in vivo* tracing studies. However, the reviewer noted that the tumor switches from glucose to glutamine metabolism for TCA cycle anaplerosis, and redox homeostasis post-radiation therapy was not a unique feature. We apologize for not clearly articulating the significance and novelty of using primary (autochthonous) mouse models of sarcoma to study *in vivo* metabolism. Recent studies suggest that the tumor microenvironment plays a significant role in determining the type of fuel that tumors use to grow. For example, experiments using typical *in vitro* culture conditions of primary lung adenocarcinoma cell lines suggested that glutamine is a predominant anaplerotic substrate for the TCA cycle. However, glutamine anaplerosis was virtually nonexistent when these cell lines were transplanted to the lung or when primary lung cancers were initiated *in vivo*. Instead, *in vivo* lung tumors use glucose-derived pyruvate as an anaplerotic substrate via the enzyme pyruvate carboxylase (PC) [3, 4]. Furthermore, in 4T1 breast tumors, PC-dependent anaplerosis was negligible, but when these tumors metastasized to the lung, PC-dependent anaplerosis increased dramatically [5]. These findings demonstrate that metabolic pathways supporting tumor growth highly depend on the tumor microenvironment in which the tumors grow and emphasize the significance of studying metabolism with *in vivo* primary models. This information (purple text above) has been added to the revised manuscript's introduction section to highlight the importance of the current study. In addition, we addressed additional concerns highlighted by Reviewer #2. Please find our response to Reviewer #2's comments in red, previous text already in the manuscript in black, and new data/text added to the manuscript in purple.

1. Fig. 1 shows that many metabolism pathways were alerted in sarcoma compared with normal tissue. Since glucose metabolism is also critical for tumor cell growth, it will be valuable if the authors could provide more information about the differences of glucose metabolism, including glycolysis and TCA cycle, between sarcoma and muscle tissue.

Reviewer #2 raises a valid point regarding the potential differential abundance of central carbon metabolites between sarcoma and normal muscle. After analyzing sarcomas versus normal muscle, we decided not to include glycolysis and the TCA cycle in the manuscript because they did not appear among the top altered pathways (Figure 1C). However, we realize that central carbon metabolism could be of interest to the readers, so we added it to the revised supplementary figure 1 with the following text in the revised manuscript:

The abundance of glycolysis and the TCA cycle metabolites were significantly higher in sarcomas compared to muscle, as shown in Supplementary figures 1A and 1B.

Supplementary figure 1: Differentially regulated metabolites in normal muscle versus sarcoma. Scatter bar graphs showed metabolite abundance in (A) Glycolysis and (B) TCA cycle pathways. p-values were calculated using multiple t-tests. Comparison of the redox and antioxidant defense state of normal muscle versus sarcoma. The ratio of (C) NAD⁺/NADH highlighting catabolism and redox state, and (D) NADP⁺/NADPH highlighting anabolism and antioxidant defense state of normal muscle versus sarcoma. p-values were calculated using t-tests. All data are represented as means \pm SD; *** p < 0.001; n = number of tissue samples.

2. The authors showed that glutamine-derived carbon flux was increased in TCA cycle in tumor cells, compared to glucose-derived carbon post-RT. However, they did not show the percentage of contribution of these two carbon suppliers in tumor cells without radiation. Thus, more experiments are needed to support the statement “irradiation of P7NP sarcomas caused a shift from glucose to glutamine dependency”.

We thank Reviewer #2 for bringing up this crucial technical point about possible missing controls for the experiment. However, we would like to draw the Reviewer’s attention toward the already present control groups in Figure 4. Infusion of ¹³C-labeled glucose and glutamine was performed 48 hours post sham (-RT) or 10 Gy RT (+RT), as depicted in Figure 4A. Solid green round dots and solid purple square dots in Figures 4C and 4D represent ¹³C-Glucose and ¹³C-Glutamine infusion in sarcoma-bearing mice without radiation (-RT), respectively. Our data revealed that sarcoma utilizes both glucose and glutamine-derived carbon for TCA cycle anaplerosis and glutathione synthesis in the absence of radiation therapy. We added the following sentence to highlight these findings in the unirradiated controls.

In the absence of radiotherapy (- RT), the incorporation of glucose- and glutamine-derived carbon into the metabolites of glutamine utilization and TCA cycle pathways were similar in sarcomas (Figure 4C, 4D).

Figure 4:

3. Numerous publications indicated that GLS expression and glutaminolysis are critical for tumor growth, but the authors showed that deletion of *Gls1* or glutaminase inhibitor treatment did not affect sarcoma tumor growth and mice survival. The authors should explain the reason for this inconsistency.

We thank Reviewer #2 for raising this point. The major difference between our studies and previously published work using glutaminase inhibitors or *Gls1* knockdown/knockout is that our experiments utilize a primary mouse model of sarcoma. For instance, Celeste Simon's group has shown that glutamine metabolism plays a crucial role in sarcomagenesis, and inhibition of glutaminase via CB-839 inhibits murine sarcoma cell growth *in vitro* and in transplant studies [6]. As described above, the tumor microenvironment plays a significant role in determining the type of fuel cancer cells utilize. Recent studies have revealed that many factors, such as genetic lesions, metabolites within tumors, pH, oxygenation, and interactions with other cells in the tumor microenvironment, play a critical role in reprogramming cancer cell metabolism [7]. However, many previous studies were performed either *in vitro* or *in vivo* using syngeneic transplant or immunodeficient xenograft tumor models, which exhibit limited inter- and intra-tumor heterogeneity and do not recapitulate gradual tumor development under immune surveillance. The mouse model of sarcoma utilized in our experiments overcomes the limitations highlighted above, and the aggressive nature of these primary tumors that have co-evolved with the immune system could be one reason for the ineffectiveness of glutaminase inhibition alone on sarcoma growth. Thus, we believe that our approach using a primary mouse model of sarcoma adds significant value in evaluating mechanisms and testing cancer drugs in preclinical settings. The sentences mentioned below have been included in the discussion section of the revised manuscript.

Understanding metabolic reprogramming in sarcomas may offer insights into tumor vulnerabilities and identify novel therapeutic targets. Recent studies have revealed that many factors, such as genetic lesions, metabolites within tumors, pH, oxygenation, and interactions with other cells in the tumor microenvironment (TME), including cancer-associated fibroblasts and immune cells, play a critical role in reprogramming the cancer cell metabolism [7]. However, many previous studies were performed either *in vitro* or *in vivo* using syngeneic transplant or immunodeficient xenograft tumor models, which exhibit limited inter- and intra-tumor heterogeneity and do not recapitulate gradual tumor development under immune surveillance. To overcome these limitations, we employed a genetically engineered mouse model of sarcoma that mimics embryonal rhabdomyosarcoma and performed both metabolomics and isotope tracing *in vivo*.

4. The authors observed increased innate immune responses in *Gls1*-deleted mice post-RT, suggesting that glutaminolysis suppressed innate immunity. Will combined treatment of glycolysis inhibitor (e.g. 2-DG) with glutaminolysis inhibitor have an additive effect on the activation of innate immunity and suppression of tumor growth?

Reviewer #2 suggests an interesting idea to increase further the radiosensitization phenotype of *Gls1*-deficient sarcomas using glycolysis inhibitors, 2-DG. Given the redundancy of metabolic pathways, blocking a single metabolic pathway has failed in clinical trials to control tumor growth, and thus, current clinical trials are evaluating a combination approach, such as targeting metabolic pathway plus RT or chemotherapy. However, targeting two major metabolic pathways by inhibiting glutamine and glucose metabolism could be toxic because normal tissues also rely on these metabolites for survival and proliferation. While we agree with the reviewer that our primary *in vivo* models would be an excellent system to investigate the potential efficacy and toxicity of combining a glutaminolysis inhibitor and glycolysis inhibitor, we feel that this proposed experiment is outside the scope of the key conclusions of the current manuscript. As the first author has recently moved to start an independent lab at Baylor and as the senior author has relocated his lab to Princess Margaret Cancer Centre, we are not in a position to complete the proposed experiment in a timely manner. This experiment can be undertaken as part of a future project.

5. “Supplementary figure 2E” on page 5 should be “Supplementary figure 1E”.

We appreciate the reviewer for pointing out this error in the main text, which we have corrected in the revised manuscript.

Reviewer #3 (Remarks to the Author):

Overall, the manuscript presents a well-written and easy-to-follow investigation of the metabolism of rhabdomyosarcoma using a murine model that carries mutations frequently observed in human embryonal rhabdomyosarcoma. The study's findings are intriguing, particularly the shift from glucose to glutamine metabolism in sarcoma cells after radiation therapy (RT) and the potential radiosensitization observed upon Gls1 deletion or pharmacological inhibition of glutaminase.

We sincerely appreciate Reviewer #3's positive comments that highlight the importance of our findings using a genetically engineered mouse model of sarcoma. To strengthen the manuscript further, we performed additional experiments and modified the text to address the concerns highlighted by the reviewer. Please find our response to Reviewer #3's comments in **red**, previous text already in the manuscript in **black**, and new data/text added to the manuscript in **purple**.

Major Comments:

1. The authors should consider including a $Gls1^{+/+}$ condition in Figures 5F-G. This addition would allow readers to compare and assess the differences more effectively between the Gls1-deficient and Gls1-normal sarcomas, providing a stronger basis for the conclusions drawn.

We appreciate Reviewer #3's suggestion to include $Gls1^{+/+}$ mice for comparison in Figure 5F-5G. The past few decades of research have highlighted the use of littermate control mice for accurate conclusions in quantitative phenotypic experiments performed using genetically engineered mouse models [8]. When conducting experiments on genetically engineered mouse strains, there is a strong preference for using littermate control mice to improve rigor and reproducibility. Thus, to study radiation response in a genetically engineered mouse model of sarcoma, we utilized $Gls1^{fl/+}$ and $Gls1^{fl/fl}$ littermate mice. In addition, we performed experiments and showed that glutaminolysis (Figure 5B), time to tumor quintupling (Figure 5C), and overall survival (Figure 5D) are similar between $Gls1^{+/+}$ and $Gls1^{fl/+}$ littermate mice bearing sarcomas. Furthermore, the glutaminase inhibitor (CB-839) study shown in Figure 5I-5J was performed using $Gls1^{+/+}$ mice. For ease of Reviewer #3 to compare the radiation response of sarcomas in $Gls1^{+/+}$ and $Gls1^{fl/+}$ mice, below we show results from vehicle-treated mice ($Gls1^{+/+}$) from Figure 5I-5J with the $Gls1^{fl/+}$ mice from Figure 5F-5G. As shown below, the radiation response measured by tumor growth and overall survival was identical between $Gls1^{+/+}$ and $Gls1^{fl/+}$ mice. Therefore, it is reasonable to retain littermate $Gls1^{fl/+}$ mice as the control for $Gls1^{fl/fl}$ mice in Figure 5F-5G in its current form.

Radiation response of vehicle-treated ($Gls1^{+/+}$) versus $Gls1^{fl/+}$ P7NP sarcoma-bearing mice. Connecting line plots showed relative tumor growth in vehicle-treated ($Gls1^{+/+}$) versus $Gls1^{fl/+}$ mice (A) without RT (- RT), and (B) with RT (+ RT). (C) Kaplan-Meier plot showed overall survival of vehicle-treated ($Gls1^{+/+}$) versus $Gls1^{fl/+}$ P7NP sarcoma-bearing mice with and without RT.

2. Figure 6, which describes the immune responses upon Gls1 deletion, appears somewhat general and does not seem to fully support the authors' conclusion regarding the increased innate immune response and changes in the sarcoma immune microenvironment. To bolster the conclusion, the authors should consider incorporating flow cytometry analysis data in Figure 6, specifically focusing on the markers used to identify the innate immune responses, and natural killer cell responses. This additional data will offer more robust

evidence for the claims made.

We appreciate Reviewer #3's suggestion to perform follow-up studies to show how the innate immune response possibly drives radiosensitization in *Gls1*-deficient sarcomas. Therefore, we performed the following flow cytometry experiment. First, we induced *Gls1*-proficient and *Gls1*-deficient P7NP sarcoma in mice by injecting 4-hydroxytamoxifen into the gastrocnemius muscle. Next, we irradiated P7NP sarcomas with 10 Gy RT, collected them 48 hours later, and dissociated them into a single-cell suspension for fluorescent antibody staining. Flow cytometry analysis revealed that natural killer (NK) cells, but not dendritic and myeloid cells, presence was significantly increased in *Gls1*-deficient compared to *Gls1*-proficient P7NP sarcomas post-RT. These findings further validate the proteomics data shown in Figure 6. The detailed method and results were incorporated in the revised manuscript to strengthen the potential role of innate immunity in radiosensitizing *Gls1*-deficient sarcomas.

Updated Method – Analyzing primary tumors by flow cytometry

To evaluate innate immune response in *Gls1*^{+/+} and *Gls1*^{fl/fl} P7NP sarcomas post-RT, we induced primary tumors using 4-OHT injection into the gastrocnemius muscle. When the tumor reached 150–400 mm³, we irradiated tumors with 10 Gy delivered using the Small Animal Radiation Research Platform or SARRP micro-CT/micro-irradiator (Xstrahl), which can deliver image-guided radiation therapy with high precision to volumes as small as 1 mm³. After 48 hours, tumor-bearing mice were sacrificed, and tumors were collected for flow cytometry. Tumors were minced and digested using a tumor dissociation kit (Miltenyi Biotec, Cat # 130-096-730) for 45 minutes on a shaker at 37°C. Dissociated tumor cells were strained through a 70 µm filter and washed with FACS buffer containing HBSS (ThermoFisher, Cat # 14175095), 0.5 mM EDTA (Sigma-Aldrich, Cat # E7889-100ML), and 2.5% FBS (Gibco, Cat # 16000044). Red blood cells were lysed using ACK lysis buffer (Lonza, Cat # 10-548E) and washed with FACS buffer. Finally, the total number of cells was counted using a hemocytometer, and 2.5M cells/tumor were used for antibody staining.

Tumor cells were blocked with 1% purified rat anti-mouse CD16/CD32 (BD biosciences, Cat # 553142) for 15 minutes at room temperature before staining with cell surface antibodies. After blocking, tumor cells were stained with APC-Cy7 conjugated anti-mouse CD45 (BD biosciences, Cat # 557659), BV711 conjugated anti-mouse CD3 (BioLegend, Cat # 100241), PE conjugated anti-mouse Nkp46 (BD biosciences, Cat # 560757), AF488 conjugated anti-mouse CD11c (BioLegend, Cat # 117313), and BV421 conjugated anti-mouse CD11b (BioLegend, Cat # 101235). Next, tumor cells were stained with Live/Dead Zombie Aqua fixable dye (BioLegend, Cat # 423101). All antibodies were used as per the company's recommendation. Finally, data were collected from ~400,000 cells/events with a BD LSRII cell analyzer and analyzed by FlowJo without the knowledge of the mouse genotype.

Updated Result – Glutaminase deletion increased the innate immune response post-radiation therapy

We performed label-free metabolomic and proteomic analyses to understand the possible mechanism(s) by which *Gls1* deletion radiosensitizes sarcomas. We irradiated *Gls1*^{+/+} and *Gls1*^{fl/fl} sarcomas with 10 Gy, sacrificed mice after 48 hours, and collected tumors to evaluate molecular alterations that might be responsible for radiosensitization. First, we analyzed the unlabeled proteomic dataset and found that overall protein abundance was significantly reduced in *Gls1*^{fl/fl} sarcomas post-RT compared to any other groups (Figure 6A, 6B). We verified that glutaminase expression at the protein level was negligible in *Gls1*^{fl/fl} sarcomas compared to *Gls1*^{+/+} sarcomas, regardless of RT status (Figure 6C). Next, we found that 490 and 217 proteins were differentially expressed in *Gls1*^{fl/fl} sarcomas post 0 Gy and 10 Gy compared to *Gls1*^{+/+} sarcomas, respectively (Supplementary figure 2). Hallmark pathway analysis revealed that *Gls1* deletion with or without RT decreased the expression of proteins related to proliferation and translation (E2F targets and G2M checkpoint, Figure 6D, 6E). Interestingly, *Gls1* deletion alone led to an immune suppressive tumor microenvironment characterized by increased cancer-associated fibroblasts (CAF) & IL2_STAT5 signaling while simultaneously decreased inflammatory & activated CD4 T cell responses (Figure 6D). Moreover, *Gls1* deletion post-RT showed an increased innate immune response marked by elevated interferon-alpha, interferon-gamma, and natural killer (NK) cell responses (Figure 6E). To validate these findings further, we performed flow cytometry to identify the presence of innate immune cells,

such as NK, dendritic, and myeloid cells, in sarcomas post-RT. As shown in Figure 6F and 6G, we irradiated an independent cohort of *Gls1*^{+/+} or *Gls1*^{fl/fl} sarcomas with 10 Gy RT, sacrificed mice after 48 hours to collect sarcomas, and dissociated them into single-cell suspension for antibody staining. Consistent with the results from proteomics analysis, we found significantly elevated levels of NK cells (Figure 6H), but not dendritic (Figure 6I) and myeloid cells (Figure 6J), in *Gls1*^{fl/fl} compared to *Gls1*^{+/+} sarcomas post-RT. The elevated numbers of myeloid cells in primary mouse sarcomas after radiation therapy was expected, given our previously published work using a different genetically engineered and carcinogen-induced mouse sarcoma model [XXXX].

Finally, we analyzed unlabeled polar and non-polar metabolites and noticed that *Gls1* deletion with or without RT caused subtle differences at the level of the metabolites (Supplementary figure 5A), but RT in *Gls1*^{+/+} sarcomas significantly reduced overall metabolite abundance compared to *Gls1*^{+/+} sarcomas (Supplementary figure 5D). In addition, we confirmed that *Gls1* deletion in sarcomas significantly reduced glutaminolysis compared to *Gls1* wildtype sarcomas, regardless of RT status (Supplementary figure 5B, 5C). Surprisingly, *Gls1* deletion with or without RT did not alter sarcomas' redox and oxidative state compared to *Gls1* wildtype sarcomas with or without RT, respectively (Supplementary figure 5E - 5H). Collectively, proteomic and flow cytometry datasets suggested that innate immune response is partly accountable for radiosensitizing *Gls1*-deficient sarcomas.

Figure 6: Glutaminase deletion increased innate immune response post-radiation therapy. (A) Heatmap showed differentially expressed proteins identified using the analysis of variance (AOV) model across treatment groups. (B) Box and whiskers plot showed associated z-scores of the relative abundance of the differentially expressed proteins. (C) Scatter bar plot showed glutaminase abundance across treatment groups. (D) Gene set enrichment of proteins that were differentially expressed in *Gls1*^{fl/fl} (0 Gy) versus *Gls1*^{+/+} (0 Gy) P7NP sarcomas. (E) Gene set enrichment of proteins that were differentially expressed in *Gls1*^{+/+} (10 Gy) versus *Gls1*^{fl/fl} (10 Gy) P7NP sarcomas. (F) Schematic representation of flow cytometry experimental design. (G) Scatter bar plot showed tumor volume at the time of sarcoma irradiation. The flow cytometry analysis revealed the presence of (H) Natural killer cells, (I) Dendritic cells,

Figure 6:

and (J) Myeloid cells in sarcoma 48 hours post-RT. Each dot in scatter bar plots represents an individual tumor sample. Associated p-values comparing z-scores across treatment groups were calculated using Wilcoxon test. p-values in scatter bar plots were calculated using multiple t-tests. All data were presented as means \pm S.D. p-values in Kaplan-Meier plots were calculated using Log-rank tests. * $p < 0.05$, ** $p < 0.01$, *** $p < 0.001$, **** $p < 0.0001$; RT – Radiation therapy; CAFs – Cancer-associated fibroblasts; EMT – Epithelial-mesenchymal transition; MDSC – Myeloid-derived suppressive cell, Gln – Glutamine, Glu – Glutamate.

3. In Figure 6, it is unclear how the authors specifically identify cancer-associated fibroblasts (CAFs) given that sarcoma cells share mesenchymal features with CAFs. It would be helpful if the authors could include markers or staining techniques used to identify CAFs in the figure, making it easier for readers to interpret the results accurately.

We thank Reviewer #3 for raising this valid point regarding the identification of cancer-associated fibroblasts (CAFs) in sarcomas. The Proteomics dataset shown in Figure 6 was computationally analyzed to identify different cell populations, including CAFs, using publicly available bioinformatic tools. The detailed information is added to the revised method section, as shown below.

Protein set enrichment analysis – We used methods developed for gene set enrichment to calculate protein set enrichment. We calculated the differential abundance between each treatment group using a linear model. The beta-coefficients from the linear model were used as the ranks into Fast Gene Set Enrichment Analysis (*fgsea*, an R library). The gene sets used in this analysis were the Hallmark Gene Sets from mSigDB (<https://www.gsea-msigdb.org/gsea/msigdb/>), immune modules from Charoentong et al. [9], and additional tumor microenvironment modules from Bagev et al. [10]

In summary, we thank all 3 Reviewers for carefully reviewing our manuscript and for providing constructive feedback. In response, we have performed additional experiments and modified the text, which we believe has further improved our manuscript.

1. Hu, Q., et al., *Genetically encoded biosensors for evaluating NAD(+)/NADH ratio in cytosolic and mitochondrial compartments*. Cell Rep Methods, 2021. **1**(7).
2. Zhang, M., et al., *HIF-1 Alpha Regulates the Response of Primary Sarcomas to Radiation Therapy through a Cell Autonomous Mechanism*. Radiat Res, 2015. **183**(6): p. 594-609.
3. Davidson, S.M., et al., *Environment Impacts the Metabolic Dependencies of Ras-Driven Non-Small Cell Lung Cancer*. Cell Metab, 2016. **23**(3): p. 517-28.
4. Sellers, K., et al., *Pyruvate carboxylase is critical for non-small-cell lung cancer proliferation*. J Clin Invest, 2015. **125**(2): p. 687-98.
5. Christen, S., et al., *Breast Cancer-Derived Lung Metastases Show Increased Pyruvate Carboxylase-Dependent Anaplerosis*. Cell Rep, 2016. **17**(3): p. 837-848.
6. Lee, P., et al., *Targeting glutamine metabolism slows soft tissue sarcoma growth*. Nat Commun, 2020. **11**(1): p. 498.
7. Schmidt, D.R., et al., *Metabolomics in cancer research and emerging applications in clinical oncology*. CA Cancer J Clin, 2021. **71**(4): p. 333-358.
8. Holmdahl, R. and B. Malissen, *The need for littermate controls*. Eur J Immunol, 2012. **42**(1): p. 45-7.
9. Charoentong, P., et al., *Pan-cancer Immunogenomic Analyses Reveal Genotype-Immunophenotype Relationships and Predictors of Response to Checkpoint Blockade*. Cell Rep, 2017. **18**(1): p. 248-262.

10. Bagaev, A., et al., *Conserved pan-cancer microenvironment subtypes predict response to immunotherapy*. *Cancer Cell*, 2021. **39**(6): p. 845-865.e7.

REVIEWERS' COMMENTS:

Reviewer #2 (Remarks to the Author):

The authors have performed additional experiments to improve the quality of this manuscript. However, some concerns remain to be addressed.

Original Point 1. The revised Fig. S1 shows glucose, glycolytic intermediates, and lactate are all decreased in sarcoma tissues compared to the muscle tissues. This result is inconsistent with the authors' statement that glycolysis is upregulated in tumor tissues.

Original Point 2. The results showed that depletion of GLS1 or treatment with CB-839 did not affect tumor growth in the model utilized in this study. The authors attributed this to the use of a primary mouse model of sarcoma, which differs from models involving transplantation. However, this explanation may not be conclusive and convincing. Yan Xiang et al. reported that the loss of one copy of GLS attenuated tumor progression in an immune-competent MYC-mediated mouse model of hepatocellular carcinoma (PMID: 25915584).

In addition, the authors used flow cytometry to analyze the profile of infiltrated immune cells in the tumor. It would be more helpful if information on CD8+ T cells and macrophages could be provided.

Reviewer #3 (Remarks to the Author):

The authors answer my comments adequately. I would suggest showing the flow-gating strategy used to generate the charts in Figure 6 (H-J), as the supplementary figure.

Reviewer #2 (Remarks to the Author):

The authors have performed additional experiments to improve the quality of this manuscript. However, some concerns remain to be addressed.

We appreciate **Reviewer #2's** additional comments noting that our additional experiments improved the quality of the manuscript further. We have addressed Reviewer 2's remaining concerns below. Please find our response to the Reviewers' comments in **red** and the new text added to the manuscript in **purple**.

Original Point 1. The revised Fig. S1 shows glucose, glycolytic intermediates, and lactate are all decreased in sarcoma tissues compared to the muscle tissues. This result is inconsistent with the authors' statement that glycolysis is upregulated in tumor tissues.

We appreciate Reviewer 2 finding an incorrect statement in the Results section. We have updated the main text in the revised manuscript to correct this error:

The abundance of the majority of metabolites in glycolysis was significantly reduced, while the abundance of the majority of metabolites in the TCA cycle was significantly higher in sarcomas compared to muscle, as shown in Supplementary figures 1a and 1b.

Original Point 2. The results showed that depletion of GLS1 or treatment with CB-839 did not affect tumor growth in the model utilized in this study. The authors attributed this to the use of a primary mouse model of sarcoma, which differs from models involving transplantation. However, this explanation may not be conclusive and convincing. Yan Xiang et al. reported that the loss of one copy of GLS attenuated tumor progression in an immune-competent MYC-mediated mouse model of hepatocellular carcinoma (PMID: 25915584).

We thank the reviewer for pointing out that haploinsufficiency of GLS attenuates tumor progression in a MYC-mediated mouse model of hepatocellular carcinoma. We have added the following sentences in the discussion section of the manuscript to address this point.

However, we found that inhibition of glutaminase alone was insufficient to impact Nras-driven primary sarcoma growth and overall survival of mice. This finding is interesting because glutaminase plays an essential role in a Myc-driven primary mouse model of hepatocellular carcinoma, where loss of one glutaminase allele significantly impacted tumor growth [1]. Taken together, these findings further support the notion that different oncogenic mutations and different tissues of origin influence nutrient dependence in cancer.

In addition, the authors used flow cytometry to analyze the profile of infiltrated immune cells in the tumor. It would be more helpful if information on CD8+ T cells and macrophages could be provided.

We appreciate that the Reviewer is interested in the impact of glutaminase inhibition and radiation therapy on CD8+ T cells and macrophages. However, our goal with the flow cytometry experiment was to understand the metabolic changes post radiation therapy (RT) and investigate how we could leverage those metabolic alterations to radiosensitize primary sarcomas. The proteomics analysis revealed that the innate immune responses, especially

natural killer (NK) cells, were implicated in radiosensitization of *G/s*-deficient sarcomas (Figure 6E). Thus, to further investigate this finding, we performed flow cytometry analysis using innate immune markers, which identified NK, dendritic, and myeloid cells in sarcomas post-RT. CD8+ T cells and macrophages were not involved in radiosensitization of *G/s*-deficient sarcomas and thus are not included in our analysis.

Reviewer #3 (Remarks to the Author):

The authors answer my comments adequately.

We appreciate **Reviewer #3's** finding that we addressed the reviewer's comments adequately. Please find our response to the Reviewers' comments in **red** and the new text added to the manuscript in **purple**.

I would suggest showing the flow-gating strategy used to generate the charts in Figure 6 (H-J), as the supplementary figure.

We thank the reviewer for this suggestion. We have explained the flow-gating strategy in the Methods section. Please see the sentences below, which appear in the revised manuscript. We believe that an additional supplementary figure won't add to this description.

During the data analysis, dead cells were first excluded using the viability dye Zombie Aqua, and then the single-cell population was isolated by plotting FSC-H vs. FSC-A. Immune cells were identified by plotting SSC-A vs. APC-Cy7-A (CD45+). After the identification of immune cells, two-parameter density plots were utilized to identify NK cells (NKp46-A vs. BV711-A), dendritic cells (AF488-A vs. BV711-A), and myeloid cells (BV412-A vs. BV711-A).

1. Xiang, Y., et al., *Targeted inhibition of tumor-specific glutaminase diminishes cell-autonomous tumorigenesis*. J Clin Invest, 2015. **125**(6): p. 2293-306.